# Adult-born neurons immature during learning are necessary for remote memory reconsolidation in rats

Marie Lods[1], Emilie Pacary [1], Wilfrid Mazier [1], Fanny Farrugia[1], Pierre Mortessagne[1], Nuria Masachs[1], Vanessa Charrier[1], Federico Massa[1], Daniela Cota [1], Guillaume Ferreira [2], Djoher Nora Abrous [1,3✉] & Sophie Tronel [1,3✉]

Memory reconsolidation, the process by which memories are again stabilized after being reactivated, has strengthened the idea that memory stabilization is a highly plastic process. To date, the molecular and cellular bases of reconsolidation have been extensively investigated particularly within the hippocampus. However, the role of adult neurogenesis in memory reconsolidation is unclear. Here, we combined functional imaging, retroviral and chemogenetic approaches in rats to tag and manipulate different populations of rat adult-born neurons. We find that both mature and immature adult-born neurons are activated by remote memory retrieval. However, only specific silencing of the adult-born neurons immature during learning impairs remote memory retrieval-induced reconsolidation. Hence, our findings show that adult-born neurons immature during learning are required for the maintenance and update of remote memory reconsolidation.

[1] University of Bordeaux, INSERM, Neurocentre Magendie, U1215, Bordeaux, France. [2] INRA, Bordeaux INP, Nutrition and Integrative Neurobiology, UMR 1286, Bordeaux Cedex, France. [3] These authors contributed equally: Djoher Nora Abrous, Sophie Tronel. ✉email: nora.abrous@inserm.fr; sophie.tronel@inserm.fr

The formation, storage and use of memories is critical for normal adaptive functioning such as problem solving, thinking, or decision making, to name a few, and can be at the center of a variety of cognitive disorders, especially in aging. For more than a century, the process by which memories are stabilized over time to become insensitive to disruption has been known as memory consolidation[1]. This process, which requires new protein synthesis, was thought to be linear, going from a labile and fragile state to a stable and permanent one. This classic view has been challenged, since we now know that consolidated memories are not permanently stable, but can again become malleable when recalled or reactivated[2–4]. Thus, a stabilized memory is not fixed, and although it can persist for a long time, it can return to a labile state. This additional process which involves de novo protein synthesis is known as memory reconsolidation and highlights that memories are highly plastic and dynamic. The great interest in the reconsolidation process is dual, as on one side it offers a window of opportunity to manipulate memories a long time after the initial encoding, and on the other, it suggests that memory reactivation may play a role in modulating memory strength and in the updating of memory content[2,5,6]. The understanding of the reconsolidation process is therefore of considerable importance to provide further insights into the development of therapeutic approaches in the treatment of pathological memories[3,7].

To date, many studies have investigated the molecular and cellular bases of both consolidation and reconsolidation[8], as well as the synaptic mechanisms underlying these processes, in particular within the hippocampus. From these studies, recent progress has been made toward finding the engram, and in particular populations of neurons that are active during memory encoding and retrieval (defined as "engram cells")[9–12]. Surprisingly, the process of reconsolidation has poorly been considered in the context of ongoing adult neurogenesis, which is known to confer new support to memory processes. Adult neurogenesis occurs in the dentate gyrus (DG) of the hippocampus. It is involved in the establishment of spatiotemporal relationships among multiple environmental cues for the flexible use of the acquired information[13–16] and helps to separate new memories from old ones by preventing interference between similar traces[17–20]. Some evidence also indicates that newborn neurons play an important role in active forgetting and memory clearance[21–23]. Furthermore, learning itself regulates adult neurogenesis[24–27]. However, it is not clear how the brain might selectively implement this process to control the dynamic or the reorganization of an established memory when reactivated or retrieved. In particular, it is not known whether different populations of neurons, at different stages of development during the initial learning, are required to maintain the memory after its reactivation.

Combining functional imaging, retroviral birth dating, and chemogenetic silencing, we examined the role of different generations of adult-born neurons in spatial memory reconsolidation. We find that remote memory retrieval activates two different populations of adult-born neurons: neurons that were immature (1–2-week old) at the time of learning and neurons that were already mature (6-week old) during learning. This remote retrieval-induced activation is absent when reconsolidation is pharmacologically blocked after prior reactivation. By chemosilencing these two populations specifically during remote retrieval-induced reconsolidation, we demonstrate that only the population that was immature at the time of learning is necessary for both the maintenance and the update of memory after its reactivation—two hallmarks of memory reconsolidation. These data reveal a functional role for adult hippocampal neurogenesis and underline a role for a population of neurons that, despite being immature at the time of learning, is yet critical for remote memory stabilization.

## Results

### Blocking protein synthesis alters remote memory reconsolidation.
We first sought to understand whether remote spatial memory could undergo reconsolidation. To this end, we followed a strategy classically used to demonstrate the existence of a reconsolidation process that consists in blocking de novo protein synthesis after memory reactivation. Rats were submitted to the water maze (MWM) training protocol for 6 days (Fig. 1a). Four weeks later, spatial memory was reactivated by submitting the animals to a probe test (Reactivation, R). During this session the platform was absent, introducing then a mismatch between the training session and the reactivation test. Immediately after reactivation, rats received a bilateral icv infusion of the protein synthesis inhibitor anisomycin (Ani-R) to block protein synthesis and thus the reconsolidation process. To be sure to manipulate a consolidated spatial information, only rats that crossed the position of the platform within the first 30 s were injected and kept in the experiment. This 30 s criterion was chosen as it has been shown that, in rats trained in the water maze, the performances at a probe test performed 30 days after training correlates with hippocampal synaptogenesis and therefore the storage of spatial representations. No such correlation is observed when the latency is over 30 s[28]. The impact of anisomycin on memory was tested 2 days later during a second probe test (Test, T). Behavioral results showed that anisomycin impaired spatial memory after reactivation (Fig. 1b, c). Latency to cross the target position was higher for Ani-R rats compared to that of control aCsf-R rats. We used another control group of rats that received anisomycin without any reactivation session to verify that the impairment of the Ani-R group was not due to anisomycin itself. Performances for the Ani group were comparable to those of aCsf-R rats. Together, these results show that blocking protein synthesis after reactivation, 4 weeks after the initial training, efficiently impairs spatial remote memory reconsolidation.

### Blocking protein synthesis alters adult-born neuron activation.
It has been shown that adult-born neurons are more prone to respond to stimuli to which they were exposed during their maturation[29] and that spatial learning influences the development of immature adult-born neurons[27]. Therefore, we first assessed whether this immature population, which is promoted by spatial learning, is activated by remote memory retrieval. We also verified whether this activation could be dependent upon reconsolidation, when the cells reached their functional maturity. To this end, rats were injected with BrdU (5-bromo-2′-deoxyuridine) 1 week before spatial training in the MWM (Fig. 1a). BrdU is a thymidine analog that has been extensively used to target proliferating cells and thus adult neurogenesis in the brain[24,30–32]. We then quantified in labeled neurons (i.e., BrdU-IR cells) expression of Zif268, a proxy for neural activity[33] critical for long-term plasticity and memory[34]. Expression of this immediate early gene in BrdU (or other analogs)-IR cells is a validated method to evaluate cell activation[35–37]. The results demonstrated that this population of adult-born neurons was activated by retrieval, but more interestingly that disrupting reconsolidation blocks this activation. The percentage of Zif268 expression in BrdU-IR cells was significantly higher in aCsf-R animals compared to that of Home Cage (HC) control and Ani-R rats (Fig. 1d, e). This blockade of activation was not due to the anisomycin infusion per se as the percentage of Zif268 in BrdU-IR cells was similar in the Ani group compared to that of aCsf-R animals and significantly higher than that of Ani-R group. To ensure that MWM

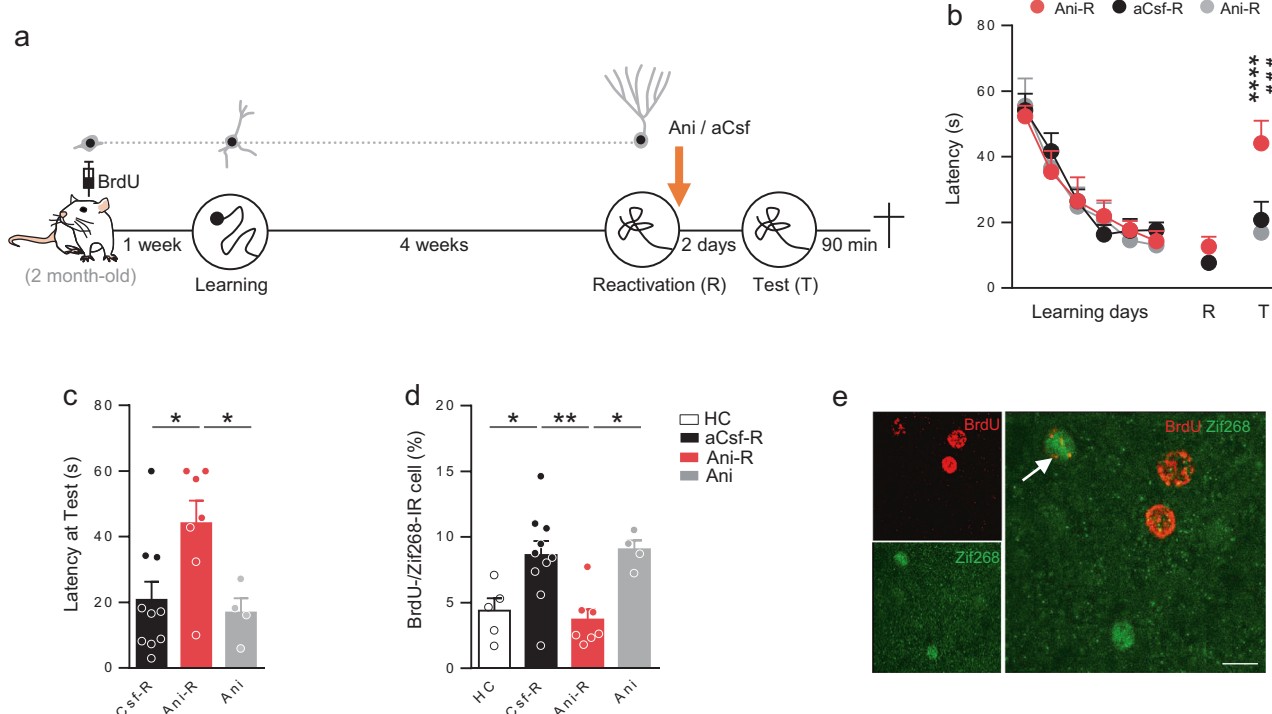

**Fig. 1 Blocking protein synthesis after spatial memory reactivation impairs both remote memory reconsolidation and adult-born neurons activation.**
**a** Experimental protocol: 2-month-old rats were injected with BrdU 1 week before MWM training. Rats were trained for 6 days and memory was reactivated 4 weeks later. Immediately after reactivation rats were injected (icv) with anisomycin (Ani-R, $n = 7$) or with aCsf (aCsf-R, $n = 10$). A group of rats received anisomycin without the reactivation session (Ani, $n = 4$). Memory was tested 2 days later and rats were sacrificed 90 min after the test. **b** Latency to find the platform during training and to first cross the position of the platform during the reactivation and test trials. Memory performances of Ani-R rats were impaired compared to those of aCsf-R rats during the test (Tukey's test: ###$p = 0.0007$) and compared to their performances at the reactivation trial (Tukey's test: ****$p < 0.0001$). **c** Latency to cross the position of the platform at the test. Latency was higher for the Ani-R rats compared to that of aCsf-R and Ani rats (Tukey's test: *$p < 0.05$). **d** Zif268 expression in BrdU-IR cells. Percentage of expression was higher in the aCsf-R group compared to that of control home cage (HC) rats ($n = 5$) and to that of Ani-R rats (Tukey's test: *$p < 0.05$, **$p < 0.01$) but not different than that of Ani rats. **e** Confocal illustration showing BrdU-IR cells (red) coexpressing the cellular activation factor Zif268-IR (green). Bar scale 10 μm. All data shown are mean ± s.e.m. For statistical details, see Table S1.

learning increases the survival of immature adult-born neurons as previously demonstrated[26,27], we analyzed the number of BrdU-IR cells in the DG. The results showed that, as expected, the number of BrdU-IR cells was higher in these groups compared to that of HC control animals (HC: 2388 ± 258; Training group: 3752 ± 183 BrdU-IR cells, $p < 0.05$). Furthermore, as shown in Fig. S1a, the number of BrdU-IR cells was not different among the learning groups. Finally, to confirm that the activation of the DG was not different between experimental groups, we analyzed the expression of Zif268 in the whole dentate gyrus. We did not find any differences among the groups (Fig. S1b). Altogether, these results demonstrate that blocking remote memory retrieval-induced reconsolidation prevents subsequent activation of neurons that were immature at the time of learning.

We then sought to determine whether the results were specific to the population of immature neurons at the time of MWM training. Taking advantage of the different BrdU analogs IdU and CldU that allow analyzing two different neuronal cohorts in the same animal[26], we targeted mature adult-born neurons and neurons born during development. Toward this end, all rats were injected with CldU 1 week after birth to label early postnatally generated neurons and with IdU at 2 months of age, as previously done[36]. Rats were trained in the MWM 6 weeks later when the developmentally-generated neurons and the adult-born neurons were mature (respectively, 13- and 6-week old). This time point was chosen as we have shown that adult-born neurons generated

6 weeks before learning are activated by both spatial learning and retrieval[38]. Memory was reactivated 4 weeks after training, as previously described. Animals received icv anisomycin immediately after reactivation, and memory was tested 2 days later (Fig. 2a). We again confirmed that blocking protein synthesis after reactivation blocks spatial remote memory reconsolidation (Fig. S2a). We then quantified Zif268 expression in both CldU and IdU-IR cells and we demonstrated that mature adult-born neurons were also activated by retrieval and that this activation was impaired when reconsolidation was blocked (Fig. 2b). The percentage of Zif268 expression in IdU-IR cells was indeed significantly higher in aCsf-R rats compared to that of HC and Ani-R animals. Again, this decrease of activation was not due to the anisomycin infusion per se since the percentage of IdU-Zif268 expression was similar in the Ani group compared to that of aCsf-R animals and significantly higher than the Ani-R group. When we analyzed the percentage of Zif268 expression in CldU-IR cells, the results showed that there was no difference among the four groups, demonstrating that early-generated neurons are not activated by remote memory retrieval (Fig. 2d, e). To ensure that the early postnatally generated cells were neurons, we analyzed the expression of Calbindin, a marker of mature neurons, in CldU-IR cells. The results showed that more than 98% of the CldU labeled cells were neurons (Fig. S3). We also showed that the whole DG activation was not different among the groups (Fig. S2b).

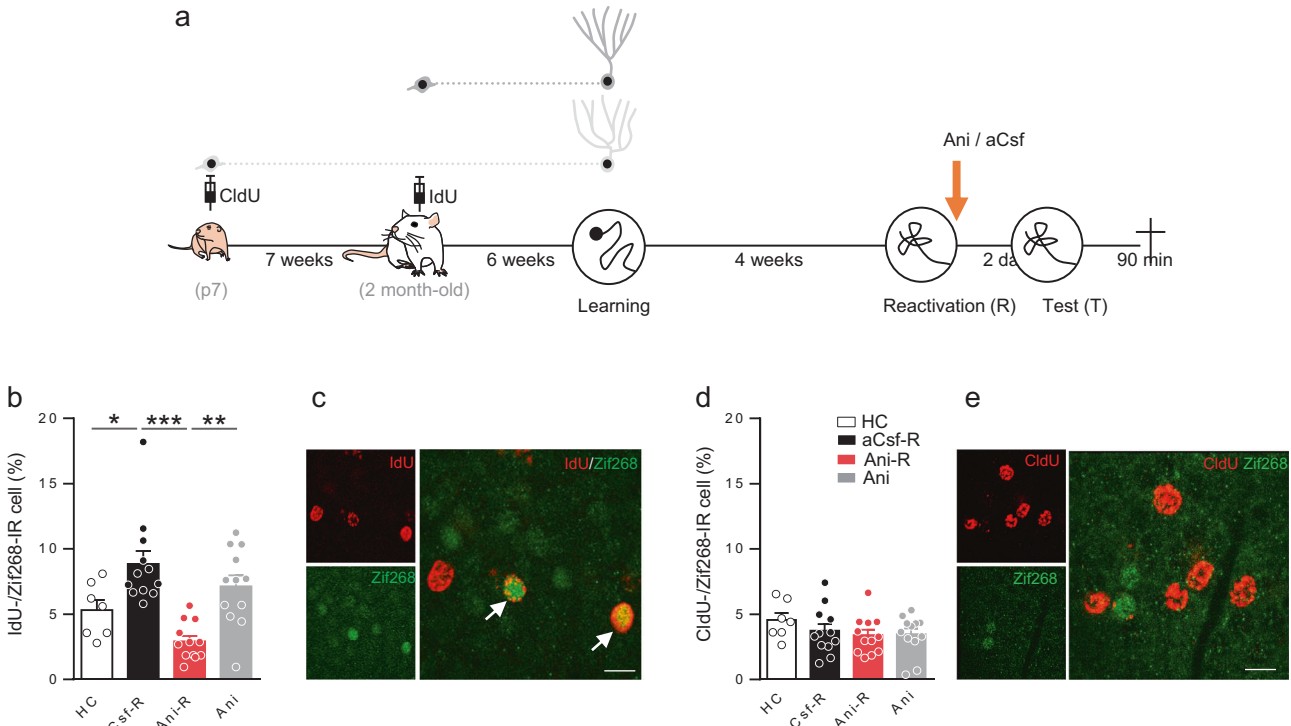

**Fig. 2 The effect of reconsolidation blockade on retrieval-induced activation is specific to adult-born neurons and not to developmentally-generated cells. a** Experimental protocol: 7-days-old rat pups were injected with CldU and later with Idu at the age of 2-month old. Six weeks later, they were trained for 6 days in the MWM and memory was reactivated 4 weeks later. Immediately after reactivation rats are injected (icv) with anisomycin (Ani-R, $n = 12$) or with aCsf (aCsf-R, $n = 12$). A group of rats received anisomycin without the reactivation session (Ani, $n = 12$). Memory was tested 2 days later and rats were sacrificed 90 min after the test. **b** Zif268 expression in IdU-IR cells. Percentage of expression was higher in the aCsf-R group compared to that of control home cage (HC) rats ($n = 7$) and to that of Ani-R rats (Tukey's test *$p < 0.05$, **$p < 0.01$, ***$p < 0.001$) but not different than that of Ani rats. **c** Confocal illustration showing IdU-IR cells (red) coexpressing the cellular activation factor Zif268-IR (green). Bar scale 10 μm. **d** Zif268 expression in CldU-IR cells. Percentage of expression was similar among the groups. **e** Confocal illustration showing CldU-IR cells (red) coexpressing the cellular activation factor Zif268-IR (green). Bar scale 10 μm. All data shown are mean ± s.e.m. For statistical details, see Table S1.

Finally, to ensure that the activation was not due to the age of the animals when the BrdU analogs were injected (2-month old in the two previous experiments), we replicated the first experiment, but this time IdU was injected in 3-month-old rats 1 week before MWM training (Fig. S4a). The results showed that the infusion of anisomycin blocked spatial memory reconsolidation (Fig. S4b) and that the retrieval-induced activation of immature neurons was inhibited when reconsolidation is blocked (Fig. S4c). As expected the number of IdU-IR cells was significantly higher in the groups of rats that were trained in the MWM (HC: 2550 ± 163; Training group: 3530 ± 234 BrdU-IR cells, $p < 0.05$) and the whole DG activation was not different among the groups (Fig. S4d). We also confirmed that the number of IdU-IR cells did not vary among trained groups (Fig. S4e).

Altogether, these results demonstrate that immature and mature adult-born neurons at the time of training are sensitive to the blockade of remote retrieval-induced reconsolidation which impedes their subsequent activation. Early postnatally generated neurons do not seem to be involved in these processes.

**Immature neurons are necessary for remote memory reconsolidation.** One question that remains to be addressed is whether adult-born neurons that are activated by remote memory retrieval are actually necessary for the reconsolidation process. One could argue that the absence of activation is only the consequence of the memory loss. In order to demonstrate a causal relationship between these different adult-born neuron populations and the process of reconsolidation, we opted to reversibly inactivate

adult-born neurons using a DREADD (designer receptor exclusively activated by designer drugs)[39] approach. For this purpose, we inserted an inhibitory DREADD (hM4-Di) construct in a GFP retrovirus (Gi-GFP-RV) which specifically transduces granular cells at their birth. This should allow their specific silencing when they will be integrated into the DG network several weeks later upon binding of the synthetic DREADD ligand Clozapine-N-oxide (CNO). First, we verified that Gi-GFP-RV infusions in the DG did not alter adult neurogenesis. Toward this end, we injected adult rats with BrdU and we bilaterally infused the Gi-GFP-RV or its control, a GFP-RV, into the DG. Rats were sacrificed 6 weeks later and the number of BrdU-IR cells was quantified. The RV infusions did not impact adult neurogenesis (Fig. S5). Then we performed whole-cell recording of Gi-GFP-RV or control GFP-RV-infected cells and assessed changes in cell excitability after CNO application (Fig. S6a). Local perfusion of CNO quickly and reversibly inhibited Gi-RV-GFP-transduced cell activity. This resulted from a decrease in both resting potential and action potential firing (Fig. S6b,c,d). No change was seen in control GFP-RV-transduced cell activity (Fig. S6e,f,g). Finally, we determined in vivo whether CNO was efficient in decreasing neuronal activation in the transduced cells. We injected rats with the Gi-GFP-RV (left hemisphere) and with the control GFP-RV (right hemisphere). Six weeks later, we injected CNO and 30 min later the convulsant pentylenetetrazol (PTZ) to increase Zif268 expression in the DG. We found that Zif268 expression in Gi-GFP-RV-transduced cells was significantly lower than that in GFP-RV infected cells (Fig. S7).

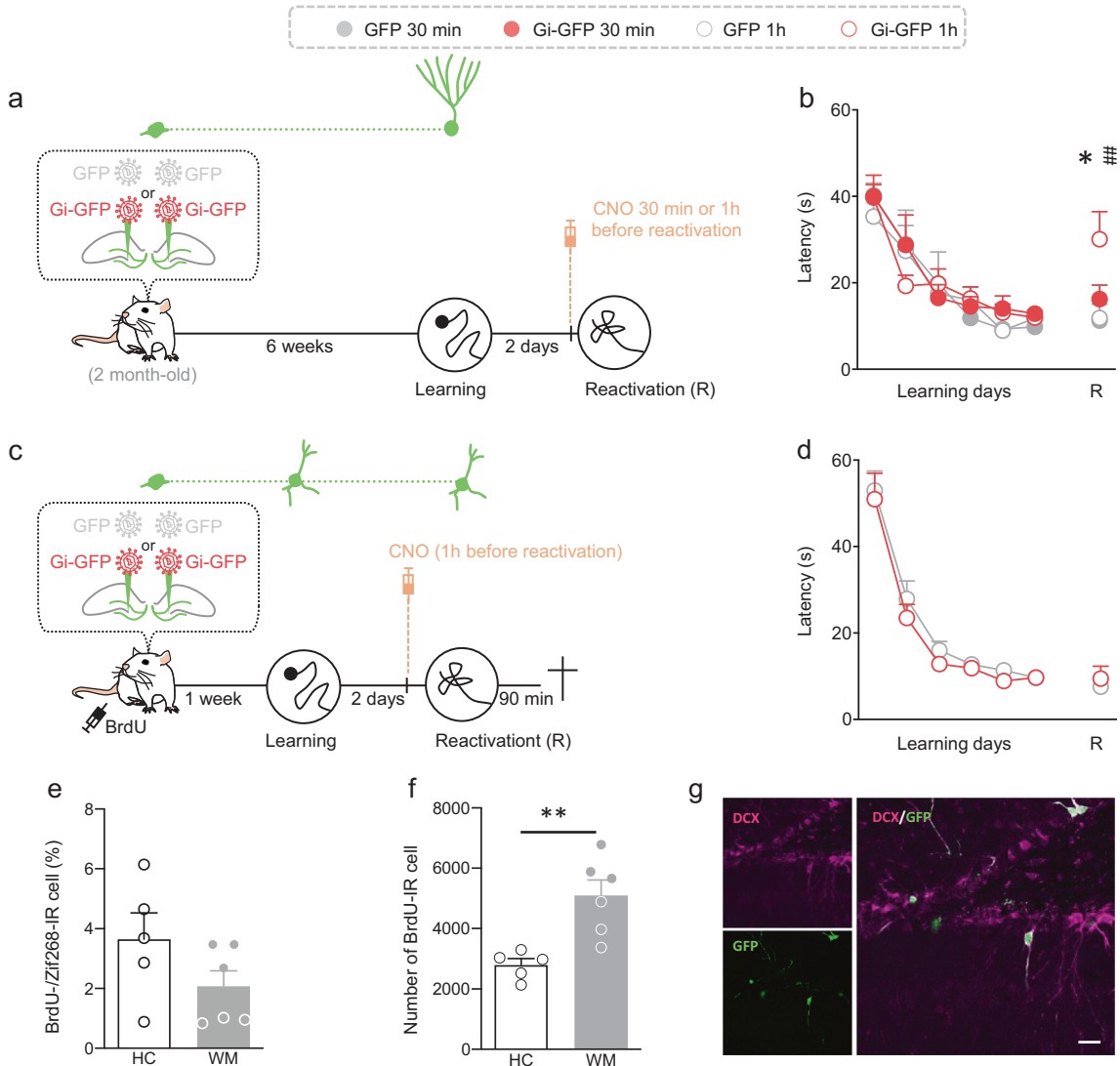

**Fig. 3 Effect of silencing immature or mature adult-born neurons on memory retrieval. a** Experimental protocol: 2-month-old rats were injected with Gi-GFP RV ($n = 20$) or its control GFP RV ($n = 14$) 1 week before MWM training. Rats were trained for 6 days and memory was reactivated 4 weeks later. Thirty minutes or 1 h before the reactivation test, CNO (1 mg/kg) was injected (i.p.) (Gi-GFP—30 min $n = 10$, Gi-GFP—1 h $n = 10$, GFP—30 min $n = 7$, GFP —1 h $n = 7$). **b** The latency to cross the platform was higher in the group of rats that received the Gi-GFP-RV and the CNO 1 h before the reactivation test compared to that of the other groups (Tukey's test: Gi-GFP—30 min vs Gi-GFP—1 h, *$p < 0.05$; Gi-GFP—1 h vs GFP—1 h, **$p < 0.01$). **c** Experimental protocol: 2-month-old rats were injected with Gi-GFP RV ($n = 10$) or its control GFP RV ($n = 7$) 1 week before MWM training. Rats were trained for 6 days and memory was reactivated 2 days later. One hour before reactivation, CNO (1 mg/kg) was injected (i.p.). Rats were killed 90 min after the test. **d** The latency to cross the platform was similar between groups. **e** The adult-born neurons were not activated as the percentage of BrdU cells expressing Zif268 in trained-GFP rats (WM, $n = 6$) was similar to that of home cage animals (HC, $n = 5$) (two-tailed $T$-test $t_9 = 1.604$, $p = 0.1433$). **f** The number of BrdU-IR cells was higher in the group of trained rats that received the GFP-RV compared to that of home cage animals (HC) (two-tailed $T$-test, $t_9 = 3.881$, $p = 0.004$). **g** Illustration of GFP-labeled cells (green) expressing DCX(magenta). Example shown is a representative of a total of >24 sections from 4 rats. Scale bar: 20 μm. All data shown are mean ± s.e.m. For statistical details, see Table S1.

Altogether, these results demonstrate that the Gi-GFP-RV does not impact adult neurogenesis development and that CNO injections can silence infected cells both ex vivo and in vivo.

Using the Gi-GFP-retrovirus, we first wanted to validate that silencing adult-born neurons by this chemogenetic approach could alter memory expression. Using an optogenetic approach, it has been shown that adult-born neurons are not necessary for spatial memory acquisition but that their reversible inactivation disrupted retrieval[40]. In order to then determine whether chemogenetic silencing of adult-born neurons impact memory, rats were injected with the Gi-GFP-RV or the GFP-RV into the DG 6 weeks before starting WM learning. Two days later, CNO (1 mg/kg) was injected i.p. in all rats one hour or 30 min before a

reactivation test (Fig. 3a). These 1 h- and 30 min-delays were chosen based on previous studies reporting that neuronal activity of DREADDs-transduced cells was optimally affected between 30 and 60 min after i.p. CNO injection[41]. As shown in Fig. 3b, memory was impaired when CNO was injected 1 h, but not 30 min, before the reactivation test. This rules out that the effect of CNO, injected 30 min before reactivation, on subsequent tests would result from a direct action on retrieval, but more likely after retrieval, i.e., during memory reconsolidation.

We then evaluated the role of immature adult-born neurons in recent memory retrieval as we have shown that ablating this population has no effect on such type of memory[42]. Rats were injected with the Gi-GFP-RV or the GFP-RV into the DG 1 week

before starting WM learning. Two days later, CNO (1 mg/kg) was injected i.p. in all rats one hour before a retrieval test (Fig. 3c). The results showed that inhibiting this population had no impact on memory expression (Fig. 3d). To determine whether this population of immature neurons was activated by retrieval, all rats received BrdU 1 week before learning. Then rats were killed 90 min after retrieval and the expression of Zif268 was analyzed in BrdU-IR cells. The results demonstrated that immature adult-born neurons are not activated since the percentage of BrdU-/Zif268-IR cells was similar in trained animals and home cage (HC) controls (Fig. 3e). As previously demonstrated the survival of immature neurons was increased by learning since the number of BrdU-IR cells was higher in the WM group compared to that of HC rats (Fig. 3f). Finally, to ensure that the Gi-GFP-RV did not transduce mature cells, we analyzed the percentage of colocalization with doublecortin, a marker of immature neurons. We found that 98% of the transduced cells were indeed immature (Fig. 3g).

Knowing that immature neurons are not necessary for recent memory retrieval we wondered whether they were necessary for recent memory reconsolidation. Toward this end, rats were injected with the Gi-GFP-RV or the GFP-RV into the DG 1 week before starting WM learning. Two days later, CNO (1 mg/kg) was injected i.p. in all rats 30 min before a reactivation test (Fig. S8a). We choose the 30-min delay for the CNO injection to be sure that the transduced cells were not inhibited during reactivation and that the CNO only affects the reconsolidation process. Then memory was tested 48 h later. As shown in Fig. S8b, silencing immature neurons does not impact recent memory reconsolidation since the performances of Gi-GFP rats were similar to those of GFP control animals.

We then determined whether the role of immature neurons evolves when the population becomes mature and integrated into the hippocampal network. Toward this end, we investigated whether the population of immature neurons at the time of learning was necessary for remote spatial memory reconsolidation. We targeted this specific population by injecting the Gi-GFP-RV or the GFP-RV into the DG 1 week before starting WM learning. Rats were also injected with BrdU (100 mg/kg). Four weeks later, when the transduced cells have reached maturity, CNO (1 mg/kg) was injected i.p. in all rats 30 min before the reactivation session. Memory was tested 2 days later (Fig. 4a). The results showed that, when memory was reactivated 4 weeks after training, latency to cross the position of the platform was similar between groups, demonstrating that CNO injections 30 min before reactivation had no impact on memory expression. Two days later, memory was tested again and the results showed that silencing during remote memory reconsolidation the population of adult-born neurons that were immature during learning, impaired subsequent memory retrieval (Fig. 4b). We next determined whether silencing the immature population at reconsolidation could have the same impact on the activation of adult-born neurons as anisomycin. To answer this question, we analyzed BrdU-Zif268 expression. The results demonstrated that this population of adult-born neurons was activated by retrieval in GFP rats and that disrupting reconsolidation in Gi-GFP animals inhibited this activation. The percentage of Zif268 expression in BrdU-IR cells was significantly higher in GFP animals compared to that of HC control and Gi-GFP rats (Fig. 4c). To confirm that the retroviruses injections had no impact on the number of BrdU-IR cells, we counted the number of BrdU-IR cells in the DG and found that there was no difference between the GFP- and the Gi-GFP rats (Fig. 4d). As expected, the survival of the immature population was increased by learning, as confirmed by the higher number of BrdU-IR cells in both trained groups compared to that of HC animals. We analyzed in the same

way the specific activation of the neurons transduced by the GFP retroviruses. We found that, similar to what we found for the BrdU-Zif268 analysis, the percentage of Zif268 expression in GFP-IR cells was significantly higher in GFP animals compared to that of HC control and Gi-GFP rats (Fig. 4f, g). Finally, to evaluate whether the effect could be due to a global decrease of the DG-CA3 network, we analyzed both DG and CA3 activation. The results showed that there was no difference between the GFP- and Gi-GFP trained groups suggesting that silencing this specific population does not lead to general disruption of the DG-CA3 circuit (Fig. S9). The estimation of transduced neurons showed that we successfully labeled 1518 ± 213 cells Gi-GFP. The analysis of the distribution of the labeled cells along the septotemporal axis of the hippocampus showed that most of the septal dentate gyrus was infected by the retrovirus (Fig. S11a,b).

We next determined whether the silencing effect was dependent upon the reconsolidation process and not due to the chemogenetic silencing during the consolidation phase. Toward this end, rats were injected with the Gi-GFP-RV or the GFP-RV into the DG 1 week before starting WM learning. Four weeks later, CNO (1 mg/kg) was injected ip in both Gi-GFP and GFP rats 30 min before the reactivation session. An additional group of Gi-GFP rats received CNO but was not submitted to the reactivation session (Fig. 5a). Memory was tested 2 days later. Again, the results showed that when memory was reactivated 4 weeks after training, latency to cross the position of the platform was similar between groups, demonstrating that CNO injections 30 min before reactivation had no impact on memory expression. Two days later, memory was tested again and the results did not reveal any difference between groups. Therefore, memory was tested again 2 weeks later. We found that long-term retention of GFP-Gi rats was significantly impaired compared to that of GFP control rats (Fig. 5b, c). When memory was not reactivated, CNO treatment had no impact on long-term retention and performances of GFP-Gi rats (GFP-Gi nR) were similar to those of GFP animals. These results suggested that silencing during reconsolidating the population of neurons that was immature at the time of leaning, disrupted long-term memory persistence. In the previous experiment (Fig. 4) all rats had good performances at the reactivation test meaning that they crossed the target position within the first 30 s. Therefore, we selected animals that have properly consolidated the spatial information, i.e., by keeping only the rats that cross the target position within the first 30 s. Interestingly, memory impairment was observed as soon as the first test session (Fig. S10a) confirming the results obtained in Fig. 4. The estimation of transduced neurons showed that we successfully labeled 1469 ± 118 cells and 1061 ± 110 cells for the Gi-GFP and Gi-GFP nR rats, respectively, with no difference between groups ($p = 0.08$) and no difference between the number of transduced cells in the previous experiment (Fig. 4); ($F_{(2,23)} = 2.164$; $p = 0.1377$). The analysis of the distribution of the labeled cells along the septotemporal axis of the hippocampus showed that most of the septal DG was infected by the retrovirus (Fig. S11a,c).

Then, we focused on mature adult-born neurons as we found that blocking reconsolidation inhibited their activation. We performed the same experiments as previously described but we injected the RV-Gi-GFP or RV-GFP 6 weeks before learning to target the mature population (Fig. 5d). In contrast to what we found previously silencing adult-born neurons during reconsolidation had no impact on the latency to reach the target zone (Fig. 5e), both if we considered all the animals or only those that performed well at the reactivation test (Fig. S10b). The number of labeled cells was similar to what we had previously obtained (1934 ± 371 cells) and the transduced cells were distributed along the septotemporal axis of the DG (Fig. S11a,d).

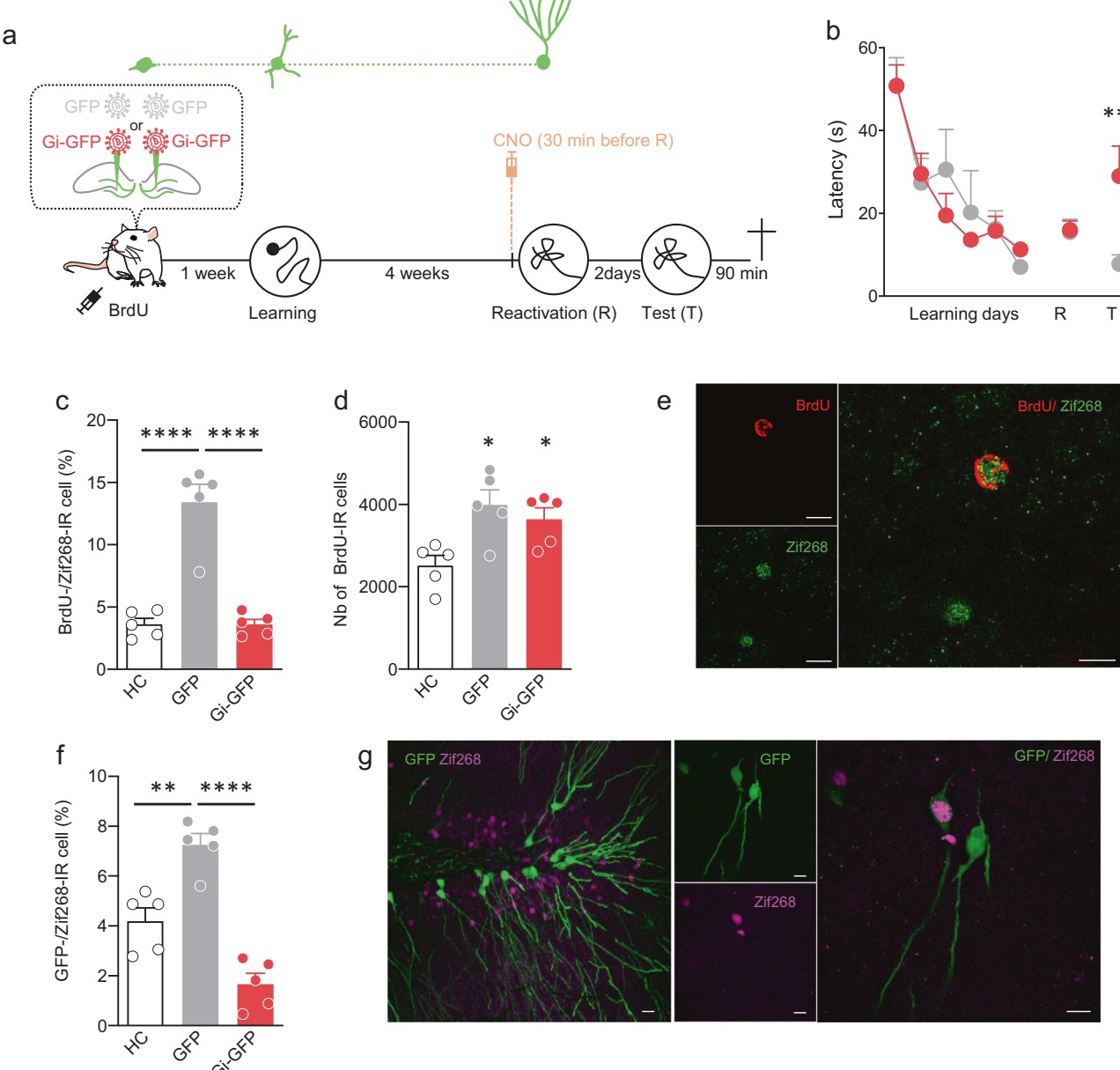

**Fig. 4 Silencing during reconsolidation, neurons that were immature at the time of learning, impairs long-term memory persistence. a** Experimental protocol: 2-month-old rats were injected with Gi-GFP RV (*n* = 5) or its control GFP RV (*n* = 6) 1 week before MWM training. Rats were trained for 6 days and memory was reactivated 4 weeks later. Thirty minutes before reactivation, CNO (1 mg/kg) was injected (i.p.). Memory was tested 2 days later (Test) and all animals were killed 90 min after the test. **b** Latency to find the platform during training and to first cross the position of the platform during the reactivation and test trial. Memory performances of Gi-GFP rats were impaired at Test compared to those of GFP RV rats (Tukey's test: **$p < 0.01$). **c** Zif268 expression in BrdU-IR cells. Percentage of expression was higher in the GFP rats compared to that of control home cage (HC, *n* = 5) rats and to that of Gi-FGP rats (Tukey's test: ****$p < 0.0001$). **d** Number of BrdU-IR cells. The number of BrdU-IR cells was similar between the GFP and Gi-GFP groups and for both higher compared to that of home cage (HC, *n* = 5) animals (Tukey's test: *$p < 0.05$). **e** Confocal illustration showing BrdU-IR cells (red) coexpressing the cellular activation factor Zif268-IR (green). Bar scale 10 μm. **f** Zif268 expression in GFP-IR cells. Percentage of expression was higher in the GFP rats compared to that of control HC rats (*n* = 5) and to that of Gi-FGP rats (Tukey's test: **$p < 0.01$, ****$p < 0.0001$). **g** Confocal illustration showing GFP-IR cells (green) coexpressing the cellular activation factor Zif268-IR (magenta). Bar scale 10 μm. All data shown are mean ± s.e.m. For statistical details, see Table S1.

Furthermore, the number of labeled cells was similar to that found when immature cells were transduced, ruling out that the difference observed between the immature and mature population in the reconsolidation process may be due to the number of transduced cells (Fig. S12a).

Finally, we wondered whether the population of neurons that were mature at the time of learning could be involved in recent memory reconsolidation. Toward this end, rats that were injected in the first Dreadds experiment (Fig. 3a, b) with the retroviruses 6 weeks before learning and injected with CNO 30 min before reactivation were tested 2 days later (Fig. S13a). As shown in Fig. S13b,c silencing adult-born neurons during reconsolidation had no impact on the latency to reach the target zone, both if we considered all the animals or only those that have correctly

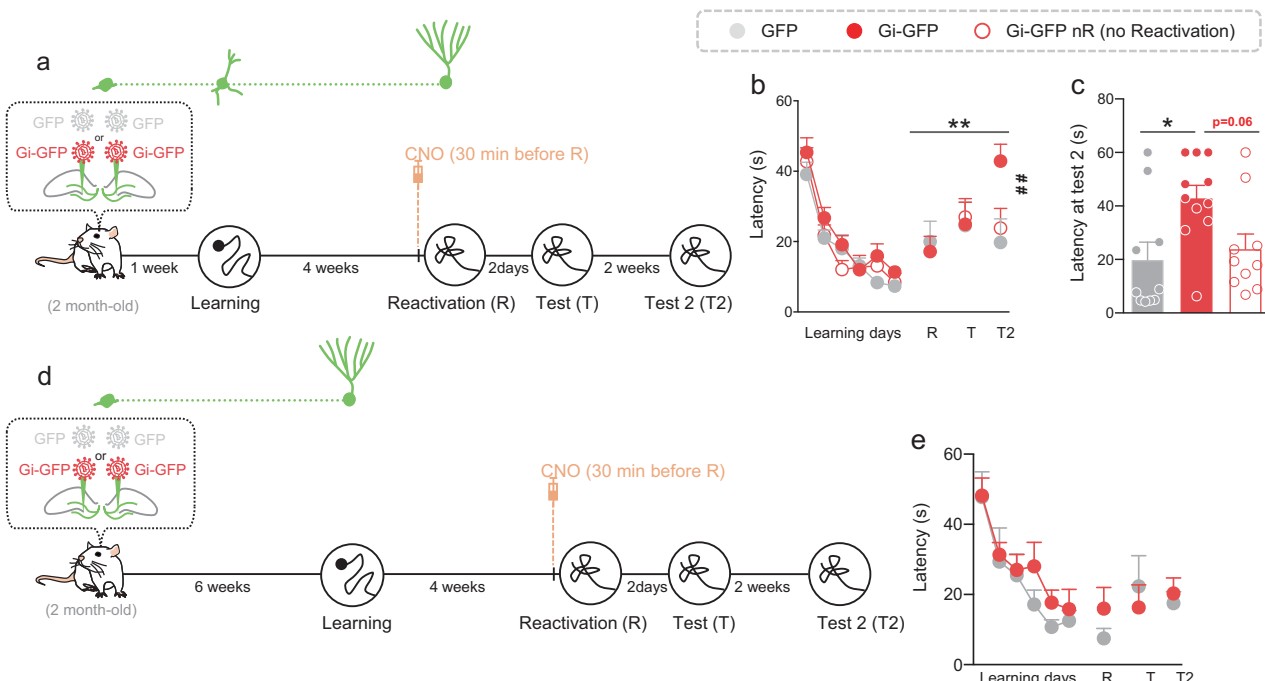

**Fig. 5 Silencing during reconsolidation, neurons that were immature at the time of learning, impairs long-term memory persistence whereas silencing neurons that were mature at the time of learning had no impact on memory. a** Experimental protocol: 2-month-old rats were injected with Gi-GFP RV ($n = 21$) or its control GFP RV ($n = 10$) 1 week before MWM training. Rats were trained for 6 days and memory was reactivated 4 weeks later GFP RV ($n = 10$) and Gi-GFP-RV ($n = 11$). Thirty minutes before reactivation, CNO (1 mg/kg) was injected (i.p.). A group of rats Gi-GFP-RV ($n = 10$) received CNO but was not reactivated (nR). Memory was tested 2 days later (Test) and again 2 weeks later (Test 2). **b** Latency to find the platform during training and to first cross the position of the platform during the reactivation and test trials. Memory performances of Gi-GFP rats were impaired at Test 2 compared to those of GFP RV rats and compared to their own performances at the reactivation trial (Tukey's test: **$p < 0.01$; ##$p < 0.01$). **c** Latency to cross the position of the platform at Test 2. Latency was higher for the Gi-GFP rats compared to that of GFP rats (Tukey's test: *$p < 0.05$). **d** Experimental protocol: 2-month-old rats were injected with Gi-GFP RV ($n = 11$) or its control GFP RV ($n = 6$) 6 weeks before MWM training. Rats were trained for 6 days and memory was reactivated 4 weeks later. 30 min before reactivation, rats were injected (i.p.) with 1 mg/kg CNO. Memory was tested 2 days later (Test) and again 2 weeks later (Test 2). **e** Latency to find the platform during training and to first cross the position of the platform during the reactivation and test trials. Memory performances of Gi-GFP rats and GFP rats are similar. All data shown are mean ± s.e.m. For statistical details, see Table S1.

consolidated the spatial information. Furthermore, when we analyzed the expression of Zif268 in BrdU-IR cells, the results demonstrated that mature adult-born neurons were activated by recent reconsolidation and that their activation was not affected by the chemogenetic silencing (Fig. S13d).

Altogether, these results suggest that, when memory is well consolidated, the inhibition of a population of neurons whose survival and development were increased by learning, disrupts long-term memory reconsolidation. Mature adult-born neurons do not seem to participate in this process.

With the passage of time, memories become generalized and the details are lost. Reconsolidation is a process by which memory can be destabilized but also by which memory can be strengthened and updated, allowing remote memory to regain its accuracy. In order to evaluate whether silencing adult-born neurons could impact a reactivation-induced update, we designed another behavioral protocol in which memory reactivation is associated with an increase accuracy of performances. Toward this end, we used an Atlantis platform, so that immediately after the 60 s reactivation session, the platform raised and rats were placed back on the platform for 30 s (Fig. 6a). By using this reactivation protocol, we observed both a significant increase in the number of entries and of the time spent in the target zone at the 48 h-test for the RV-GFP control rats (Fig. 6b–d). However, inhibiting the population of neurons that was immature at the time of learning disrupted this reactivation-induced increase of performances. At the end of the 48 h-test, the Atlantis platform

raised and the rats were placed 30 s on it again. Two weeks later the Gi-GFP rats still performed worse than the GFP rats. The Gallagher index[43] was calculated to estimate the average distance of rats from the former platform (proximity) as well as the efficiency index. Both indexes demonstrated that control GFP rats had improved performances, as their search paths are closer to the former platform and more efficient reaching the target zone (Fig. 6e–g). The estimation of infected neurons showed that we successfully labeled 1871 ± 228 cells and that the labeled cells were distributed along the septotemporal axis of the DG (Fig. S11a,e). One could argue that this reactivation trial could induce a new learning. However, if it was the case we should see an improvement in the accuracy of memory between test T and test T2 since animals are also submitted to the update-reactivation at test T. This was not the case, suggesting that the initial trace was disrupted rather than a new learning. The latency to cross the platform was not disrupted (Fig. S14a) which could be explained by the fact that the reinforced memory was too strong and therefore not sensitive to disruption[44]. Only the increase of memory accuracy promoted by the reactivation-induced update was disrupted.

Altogether these results demonstrated that blocking the neurons that were immature during learning disrupt memory updating induced by reconsolidation.

Finally, when we used the Atlantis protocol to investigate the involvement of the population of neurons that was mature at the time of learning (Fig. 7a), we observed in both GFP- and Gi-GFP

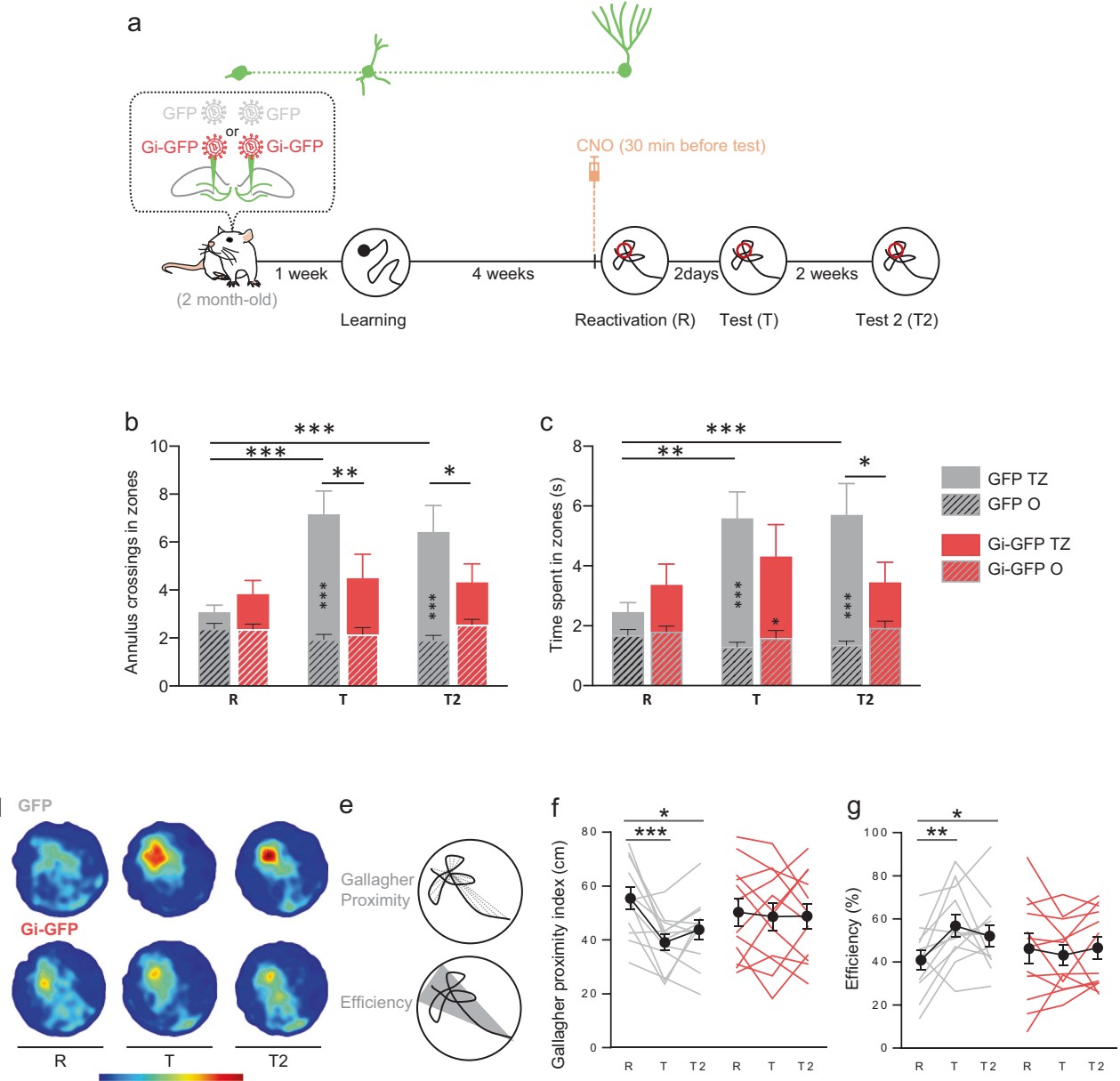

**Fig. 6 Silencing during reconsolidation, neurons that were immature at the time of learning, impairs memory update. a** Experimental protocol: 2-month-old rats were injected with Gi-GFP RV ($n = 12$) or its control GFP RV ($n = 12$) 1 week before MWM training. Rats were trained for 6 days and memory was reactivated 4 weeks later. Thirty minutes before reactivation, rats were injected (i.p.) with 1 mg/kg CNO. Memory was tested 2 days later (Test) and again 2 weeks later (Test 2). At the end of each reactivation and test sessions, the Atlantis platform (indicated by ◯) raised and rats were put on the platform for 30 s. **b**, **c** Cross entries and time spent in the MWM zones (T: Target zone; O: others zones). The reactivation with the Atlantis platform led to an increase in the number of entries and the time spent in the target zone at the test for the GFP control rats at test T and T2 (Tukey's test: $*p < 0.05$, $**p < 0.01$, $***p < 0.001$). Atlantis reactivation had no effect on Gi-GFP performances at the tests. GFP rats enter more and spent more time in the target zone during tests than Gi-GFP rats (Tukey's test: $*p < 0.05$, $**p < 0.01$). **d** Density plot for grouped data: The color level represents the lowest (Min) to the highest (Max) location frequency in pixels. **e** Gallagher Proximity: Average distance of rats from the former platform during the first 20 s. Efficiency: Percentage of distance crossed in the represented triangular zone. **f**, **g** Gallagher Proximity and Efficiency during reactivation and tests. Atlantis reactivation led to an increase in Gallagher and Efficiency performances at the tests for the GFP control rats (Tukey's test: $*p < 0.05$, $**p < 0.01$, $***p < 0.001$). Atlantis reactivation had no effect on Gi-GFP performances at the tests. All data shown are mean ± s. e.m. For statistical details, see Table S1.

rats an improvement in performance in the number of entries (Fig. 7b) and of the time spent in the target zone (Fig. 7c, d), as well as in the proximity and efficiency indexes (Fig. 7e, f). As expected, the latency to cross the platform was not disrupted (Fig. S14b). This was not due to the number of labeled cells that was similar to what we had previously obtained (1648 ± 321 cells) and

that the labeled cells were distributed along the septotemporal axis of the DG (Fig. S11a,f). Once again, the number of labeled cells was similar to that found when immature cells were transduced, ruling out that the difference observed between the immature and mature population in the reconsolidation process may be due to the number of transduced cells (Fig. S12b).

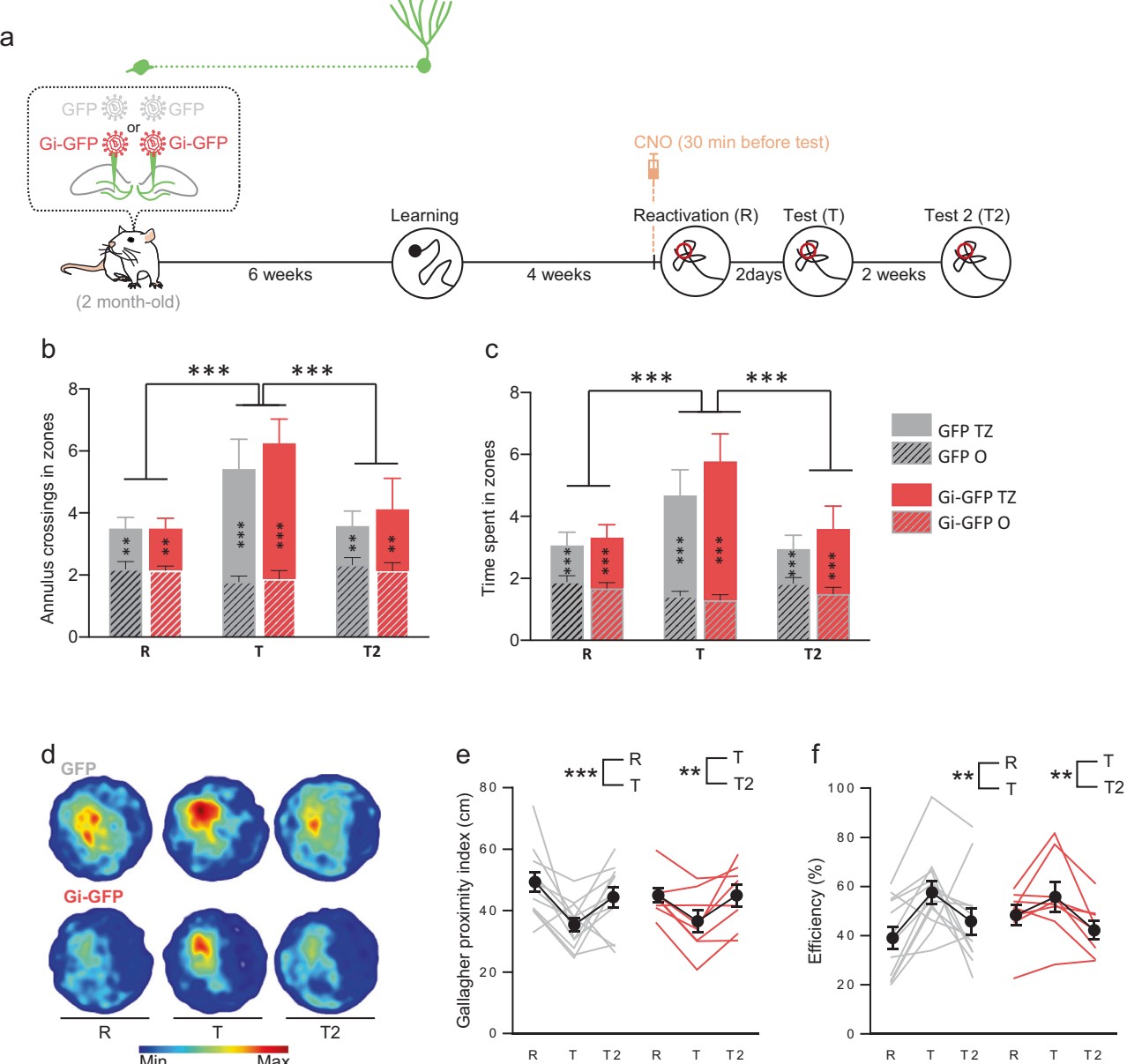

**Fig. 7 Silencing during reconsolidation, neurons that were 6-week old at the time of learning, has no impact on memory updating. a** Experimental protocol: 2-month-old rats were injected with Gi-GFP RV (n = 12) or its control GFP RV (n = 11) 6 weeks before MWM training. Rats were trained for 6 days and memory was reactivated 4 weeks later. Thirty minutes before reactivation, rats were injected (i.p.) with 1 mg/kg CNO. Memory was tested 2 days later (Test) and again 2 weeks later (Test 2). At the end of each reactivation and test sessions, the Atlantis platform (indicated by ○) raised and rats were put on the platform for 30 s. **b, c** Cross entries and time spent in the MWM zones (T: Target zone; O: others zones). The reactivation with the Atlantis platform led to an increase in the number of entries and the time spent in the target zone at the test for both GFP control rats and Gi-GFP rats (Tukey's test: **p < 0.01, ***p < 0.001). **d** Density plot for grouped data: The color level represents the lowest (Min) to the highest (Max) location frequency in pixels. **e, f** Gallagher Proximity and Efficiency during reactivation and tests. The reactivation with the Atlantis platform led to an increase in Gallagher and Efficiency performances at the test for both GFP control rats and Gi-GFP rats (Tukey's test: **p < 0.01, ***p < 0.001). All data shown are mean ± s.e.m. For statistical details, see Table S1.

These results suggest that silencing the adult-born neurons population that was mature at the time of learning does not impair long-term memory reconsolidation.

## Discussion

Memory reconsolidation has been extensively studied over the past decades in all sort of behavioral paradigms but mostly on pavlovian conditionings[45]. Very few studies have focused on spatial memory reconsolidation[44,46–52] and, to our knowledge, none has investigated remote spatial memory reconsolidation. In

agreement with previous studies[46], we found that reconsolidation occurs when reactivation introduces a mismatch between the training session and the reactivation test. The fact that the platform was absent at the reactivation trial induced a discrepancy between actual and expected events, i.e., a prediction error[53–55]. This mismatch also occurs in our Atlantis protocol since the platform was absent from the pool in the first 60 s of the test.

In addition to this finding, we have demonstrated that 4 weeks after training, spatial remote memory can be destabilized by anisomycin injection after reactivation. This finding contrasts

with results obtained using other memory tasks, such as inhibitory avoidance, which becomes less susceptible to undergo reconsolidation with the passage of time[56]. This could be related to the fact that the hippocampus is involved in spatial memory regardless of the time elapsed since learning, and therefore these results add another stone to the edifice of the boundary conditions for reconsolidation. A similar effect was also observed by silencing the population of adult-born neurons that was immature at the time of learning, demonstrating a new role for adult neurogenesis in the long-term memory reconsolidation.

Previous studies provided evidence that mature adult-born neurons are activated and required during the formation and the expression of spatial memory[36,38]. Furthermore, memory formation is associated with an expansion of their dendrites and spines[42] that most probably strengthens synaptic connections with their inputs from local or extrahippocampal areas. Spatial memory depends upon to this network as post-training ablation of mature adult-born neurons impairs memory expression[57]. In contrast, many studies have shown that when immature, new neurons are not required for spatial learning[15,42,58]. However, thanks to learning-induced acceleration of their development and maturation and by the early arrival of glutamatergic inputs[27,59] they are, as they age, activated by memory expression[60].

In our experiments, we found that adult-born neurons are activated during the post-reactivation test (T) as shown by the increase in the percentage of Brdu or IdU cells expressing Zif268 during the post-reconsolidation test (T). This process is blocked by inhibition of protein synthesis indicating that independently of their age at the time of learning, adult-born neurons' activation is sensitive to disruption of retrieval-induced reconsolidation. This activation is specific to the neurons born during adulthood, at least up to 3-month-old rats, since the percentage of developmentally-generated cells was not affected by remote retrieval or blockade of retrieval-induced reconsolidation.

Surprisingly, although both immature and mature adult-born neurons are activated by remote retrieval, only the population of neurons that was immature at the time of learning seems to participate in remote memory reconsolidation. In fact, silencing this specific population and not the population that was mature during training impairs reconsolidation as revealed by impaired subsequent memory expression. The effects are persisting until 2 weeks after the treatment, suggesting that silencing immature neurons during the reconsolidation process has enduring effects on the memory trace. The fact that mature adult-born neurons are not required for the process of reconsolidation suggests that different adult-born populations could be involved in retrieval and in its resulting process, i.e., reconsolidation, in the same way that different molecular mechanisms can be involved in the destabilization of memory induced by retrieval and the following restabilization allowing reconsolidation process[50]. Therefore, our results are in agreement with previous studies showing that post-training ablation of adult-born neurons that were mature at the time of training are required for the expression of memory measured during a probe test[57]. Furthermore, we demonstrated that the involvement of immature adult-born neurons in reconsolidation was specific to remote memories since recent reconsolidation is not affected by immature nor mature adult-born neurons silencing. These results reinforce our conclusions that only those immature neurons whose development and maturation have been influenced by learning are required to stabilize or reinforce remote memories after reactivation.

It should be noted that our results slightly disagree with those obtained by Suarez-Pereira and Carrion[61]. In their study, the authors reported that immature adult-born neurons are involved in reconsolidation of object recognition memory. In contrast to what we found, adult-born neurons were still immature when their ablation affected the process of reconsolidation. These discrepancies may be due to differences in the manipulation used (ablation versus transient silencing) or the cognitive paradigms used. However, Suarez-Pereira and Carrion demonstrated that immature adult-born neurons express the immediate early gene Zif268 after training or after reactivation which is in sharp contrast with our study. Therefore, the population that they targeted is already activated in these processes which is not the case in our study which could explain the discrepancies. Indeed, we showed that the immature population is not activated by recent spatial memory reactivation/retrieval (Fig. 3e).

Altogether, these results suggest that mature adult-born neurons could sustain encoding and memory expression whereas immature ones at the time of learning would be involved in the stabilization of the trace after behavioral activation. This highlights the role of the population of neurons that are immature at the time of learning and whose development and maturation are accelerated by spatial learning. Moreover, since its silencing upon reconsolidation impacts long-term stabilization of the memory trace, we could propose that this population can act on the network recruited during spatial memory formation, i.e., the engram[11]. One question that remains to be answered is how this immature population could influence this memory engram network. It was shown that immature neurons are able, under certain conditions, to display synaptically driven action potentials[62], although with a lower firing rate than mature ones. In addition, this population is shaped by spatial learning indicating that it is able to integrate stimuli generated in the course of learning[27]. It could be speculated that they are primed during learning and, once mature, they influence the neural representation of the learned information as it has been shown in olfactory memory[63]. Thus, they could modulate memory when reaching full functionality as they age. What could be the mechanisms involved? First, it has been shown that an accelerated integration of immature neurons triggered by enriched environment is dependent on parvalbumin neurons[64]. This might speed up the functional significance of immature neurons by expanding their connectivity. When mature, they might then be fully integrated into the network and thus they become necessary to the reconsolidation of the information to which they were exposed during their early maturation. Furthermore, it has been shown using in vivo calcium imaging that young adult-born neurons (3–6-week age range) are more active compared to older cells[65], a finding consistent with the observation that they more excitable ex vivo[62]. This higher excitability together with a lower threshold for plasticity may render them less selective in terms of response to the activity driven by learning and thus less spatially tuned than the mature population[66]. This could explain why new neurons do not encode spatial information when they are young but will represent the events as they mature.

The reason why mature adult-born neurons are not required for reconsolidation is unclear. This assembly is activated during memory reconsolidation but silencing the cohort of mature neurons does not impact this process. Two non-exclusive explanations are conceivable: memory expression is supported by the remaining non-infected adult-born neurons or by the population that was immature at the time of learning. Given that silencing the immature neurons was sufficient to disrupt reconsolidation, it can be proposed that they win the competition because they are more prone to be activated by an experience that shaped their development. In contrast, the likelihood of modifying the trace by mature neurons that have encoded unique features during learning is decreased.

In conclusion, adult neurogenesis seems to play a critical role in established reactivated memories. In addition to their role in learning, forgetting, and pattern separation[1], we suggest that the

role of adult-born neurons in reconsolidation is dependent on their maturation state, but more importantly experience-induced plasticity they have encountered at different stages of their integration.

## Methods

**Animals**. A total of 297 male Sprague-Dawleys rats (OFA, Charles River, France) were used for these experiments.

Rats weighing between 250 and 275 g (2 months of age) at the time of delivery were individually housed in standard cages under a 12/12 h light/dark cycle with ad libitum access to food and water.

Pregnant female ($n = 10$, 3 months; 240–260 g body weight on delivery) Sprague–Dawley rats (OFA, Charles Rivers, France) were individually housed in plastic breeding cages under standard laboratory conditions (12 h/12 h light/dark cycle, 22 °C, 60% humidity, water and food available ad libitum). After birth, only litters of 8–13 pups with approximately equal sex-ratios were retained for the study. The litters were raised by their biological mothers until weaning (21 days after birth). After weaning, only the male progeny was kept, and animals were randomly assigned to the different experimental groups.

All experiments were performed in accordance with the recommendations of the European Union (2010/63/UE) and were approved by the ethical committee of the University of Bordeaux (#Dir1367; #Dir23375).

**Plasmids and retroviruses**. The Gi DREADD was cloned by PCR using pcDNA5/FRT-HA-hM4D(Gi) (Addgene #45548[67]) as a template (see Table S2 for PCR primers) and then inserted into the BamHI site of a CAG-IRES-GFP retroviral backbone[68]. The resulting construct CAG-Gi-IRES-GFP was sequenced using specific primers (Table S2) and is named Gi-GFP-RV in the text. The control construct had the same viral backbone without the insert (GFP-RV in the text).

High titers of retroviruses were prepared with a human 293-derived retroviral packaging cell line (293GPG)[69], kindly provided by Dr. Dieter Chichung Lie (University of Erlangen-Nuremberg). Virus-containing supernatant was harvested 3 days after transfection with Lipofectamine 2000 (Invitrogen, Oregon, USA. #11668-019). This supernatant was then cleared from cell debris by centrifugation at $2191 \times g$ for 15 min and filtered through a 0.45 µm filter (Millipore, Massachusetts, USA). Viruses were concentrated by two rounds of centrifugation (respectively, $46{,}000 \times g$ and $67{,}629 \times g$, 1 h each) and resuspended in PBS.

**Injection of thymidine analogs**. In the first experiment, adult rats aged of 2 months were injected with BrdU (5-bromo-2′-deoxyuridine, 100 mg/kg). In the second experiment, adult rats, respectively, aged 3 months and 1 week were injected with IdU (5-Iodo-2′-deoxyuridine, 100 mg/kg). In the third experiment, pups were injected at postnatal week 1 (PN 1 W) with CldU (5-chloro-2′-deoxyuridine; $1 \times 50$ mg/kg) and at 2 months old with IdU (100 mg/kg). BrdU, IdU, and CldU were dissolved in Phosphate Buffer (pH 8.4), 1 N NH$_4$OH/NaCl and NaCl, respectively.

**Water maze procedures**. The apparatus consisted of a circular plastic swimming pool (180-cm diameter, 60-cm height) that was filled with water ($20 \pm 1$ °C), rendered opaque by the addition of a white cosmetic adjuvant. Two days before training, the animals were habituated to the pool for 1 min. During training, animals were required to locate a submerged platform (16-cm diameter) hidden 1.5 cm under the surface of the water in a fixed location, using the spatial cues available within the room. All rats were trained for four trials per day (90 s with an inter-trial interval of 30 s and released from three starting points used in a pseudorandom sequence each day) during 6 days. If an animal failed to locate the platform, it was placed on that platform at the end of the trial. The time to reach the platform was recorded with a video camera that was fixed to the ceiling of the room and connected to a computerized tracking system (Videotrack, Viewpoint, Lyon, France) located in an adjacent room.

*Classical reactivation protocol*. Four weeks after learning, rats were submitted to a reactivation test in the water maze. During this session, rats were put 60 s in the water maze without the platform. Performances were assessed using latency to cross the position where the platform was during acquisition.

*Probe test*. Two days after reactivation, all rats were tested in the water maze for 60 s.

*Atlantis reactivation protocol*. Four weeks after learning, rats were submitted to a reactivation test in the water maze. During this session, rats were put 60 s in the water maze with the Atlantis platform maintained at the bottom of the pool. After 60 s, the platform raised automatically and rats were placed onto it for 30 s. Performances were assessed using several parameters: Latency to cross the position where the platform was during acquisition and and the amount of time spent and the number of entries (annulus crossing) done in each zone. Zones were defined as an ideal circle (30-cm diameter) located at the original platform location (Target zone; T) and the three equivalent areas in each other quadrants (other zones; O).

Gallagher index: Average distance in centimeters of rat from the center of the platform location across the first 20 s of test. Efficiency index: distance traveled in an ideal triangular zone with an angle at the starting point and the opposite base at the platform level across the first 20 s of test (POLY File; Imetronic).

**Surgery**. Rats were anaesthetized with 3% isoflurane and placed in the stereotaxic frame, where they were maintained on 2% isoflurane for the duration of the surgery. Analgesia was provided by a subcutaneous injection of Metacam (1 mg/kg).

*Cannula implantation*. Stainless steel cannulas (28 gauge) were stereotaxically implanted bilaterally into the lateral ventricles (−1.3 mm posterior to Bregma, 1.8 mm lateral from midline, and 3.4 mm ventral). After surgery, rats were returned to their home cage for a 7-d recovery period. At the end of the experiment, cannula implantation was checked and rats with cannulas that were not correctly located in the lateral ventricles were discarded from the analysis.

For retroviral injections, retroviruses were stereotaxically injected (2 µL per injection site at 0.3 µL/min) into the dentate gyrus of adult rats with a microcapillary pipette connected to a micro-syringe pump (KDScientific SPLG130) attached to the stereotaxic frame. Four bilateral injections were made for behavioral and electrophysiological experiments (−3.2 mm posterior, ±1.6 mm lateral, −4.2 ventral; and −3.8 mm posterior, ±1.8 mm lateral, −4.2 mm ventral). At the end of the experiment, only rats with labeled cells in both hemispheres were kept in the analysis.

**Anisomycin infusion**. Anisomycin (A9789, Sigma-Aldrich) was dissolved at the dose of 125 µg/µl[70]; 3 µl per side were infused into the lateral ventricles at a rate of 1 µl/min.

**CNO delivery**. The Dreadd ligand CNO (Clozapin-n-Oxyde, Enzo Life Sciences, Lyon, France. #BML-NS105) was dissolved in a saline solution and delivered with one i.p. injection of 1 mg/kg in rats 30 min or 1 h before the test.

**Immunohistochemistry and analysis**. Animals were perfused transcardially with a phosphate-buffered solution of 4% paraformaldehyde. After 1 week of fixation, brains were cut with a vibratome. Free-floating 50-µm-thick sections were processed according to a standard immunohistochemical procedure to visualize GFP (Chicken primary antibody, 1:2000, Abcam, Cambridge. UK. #Ab13970), BrdU (Mouse primary antibody, 1:200, Dako Agilent, Santa Clara, USA. #M0744), IdU (Mouse primary antibody, 1:500, BD #347580), CldU (rat primary antibody, 1/1000, Accurate C&S CO), and Zif268 (Rabbit primary antibody, 1:500, Santa Cruz Biotechnology, Santa Cruz, USA. #SC-189) on alternate 1-in-10 sections[4].

GFP positive cells throughout the entire dentate gyrus were revealed using the biotin-streptavidin technique (ABC kit, Vector Labs, Peterborough, UK #PK-4000) and 3,3′-diaminobenzidine as a chromogen with a biotinylated goat anti-chicken antibody (1:1000, Jackson ImmunoResearch, Cambridgeshire, UK. #103-065-155).

GFP-IR cells were counted under a ×100 microscope objective throughout the entire septotemporal axis of the granule and subgranular layers of the dentate gyrus (DG). The total number of cells was estimated using the optical fractionator method, and the resulting numbers were tallied and multiplied by the inverse of the sections sampling fraction (1/ssf10).

XdU-positive cells throughout the entire granular layer of the supragranular and infragranular blades of the DG were revealed using the biotin-streptavidin technique with a horse anti-mouse for BrdU and IdU (1:200, Vector Labs, Peterborough, UK. #BA-2001) antibody and with a goat anti-rat for CldU (1:200, Vector Labs, Peterborough, UK. #BA-9400) antibody. The total number of cells was counted under a ×100 microscope objective throughout the entire left septotemporal axis of the granule and subgranular layers of the DG. The total number of cells was estimated using the optical fractionator method.

To estimate the density of Zif268-IR cells (both sides), two-dimensional images of the entire dentate gyrus and CA3 were acquired with a slide scanner. The slide scanner was a Nanozoomer 2.0HT with a fluorescence imaging module (Hamamatsu Photonics France) using an objective UPS APO ×20 NA 0.75 combined to an additional lens ×1.75, leading to a final magnification of ×35. Virtual slides were acquired with a TDI-3CCD camera. Zif268-IR cells were counted by using Mercator software (Explora Nova). The results are expressed as the number of IR cells per mm$^2$ of the granule cell layer of the DG and the CA3.

Activation of XdU-IR neurons was examined by using immunohistofluorescence. Sections were incubated with BrdU antibodies from different vendors (Brdu and CldU, 1/500, Accurate Chemical and Scientific Corporation; IdU, 1/500, BD Bioscience). Bound antibodies were visualized with Cy3-goat anti-rat antibodies (1/1000, Jackson for CldU), or Cy3-goat anti-mouse antibodies (1/1000, Jackson for IdU). Sections were also incubated with Zif268 rabbit (1/500, Santa Cruz Biotechnology) antibody. Bound antibody was visualized with Alexa-488-goat anti-rabbit antibody (1/1000, Invitrogen) or Cy5-goat-anti-rabbit antibody (1/1000 Chemicon). Primary antibodies for CldU or IdU and Zif268 were incubated simultaneously at 4 °C for 72 h, and secondary antibodies were incubated simultaneously at RT for 2 h.

For GFP/DCX or GFP/Zif268 colocalization, sections were incubated with GFP antibody (Chicken primary antibody, 1:1000, Abcam, Cambridge, UK. #Ab13970)

and DCX antibody (Rabbit primary antibody, 1/100, Sigma-Aldrich) or Zif268 antibody (1/500, Santa Cruz Biotechnology).

For CldU/Calbindin colocalization, sections were incubated with CldU (1/500, Accurate Chemical and Scientific Corporation) and Calbindin (1/125, Santa Cruz Biotechnology, Santa Cruz, USA. #SC-7691). Bound antibodies were visualized with Cy3-goat anti-rat antibodies (1/1000, Jackson for CldU) and A647-Donkey anti-goat antibodies (1/1000, Jackson for Calbindin).

Double labeling was determined by using a SPE confocal system with a plane apochromatic ×63 oil lens (numerical aperture 1.4; Leica) and a digital zoom of 2. The percentage of XdU cells expressing IEG (all along the temporal–septal axis) was calculated as follows: (Nb of XdU$^+$-IEG$^+$ cells)/(Nb of XdU$^+$-IEG$^-$ cells + Nb of XdU$^+$-IEG$^+$ cells) × 100. In all experiments, a minimum of 200 adult-born cells were analyzed per rat.

**Electrophysiological recordings.** Animals were deeply anesthetized (167 mg/kg ketamine and 16.7 mg/kg xylazine) and sacrificed. Dissected brain was immediately immerged in ice-cold oxygenated cutting solution (in mM: 180 Sucrose, 26 NaHCO$_3$, 11 Glucose, 2.5 KCl, 1.25 NaH$_2$PO$_4$, 12 MgSO$_4$, 0.2 CaCl$_2$, saturated with 95% O$_2$–5% CO$_2$). Three-hundred-and-fifty-micrometer slices were obtained using a vibratome (VT1200S Leica, Germany) and transferred into a 34 °C bath of oxygenated aCSF (in mM: 123 NaCl, 26 NaHCO$_3$, 11 Glucose, 2.5 KCl, 1.25 NaH$_2$PO$_4$, 1.3 MgSO$_4$, 2.5 CaCl$_2$; osmolarity 310 mOsm/l, pH 7.4) for 30 min and then cooled down progressively till room temperature (RT; 23–25 °C) in oxygenated aCSF. After a 45-min recovery period at RT, slices were anchored with platinum wire at the bottom of the recording chamber and continuously bathed in oxygenated aCSF (RT; 2 ml/min) during recording.

Transduced newborn granular cells were identified using GFP with a fluorescence/infrared light (pE-2 CoolLED excitation system, UK). Neurons action potential firing was monitored in whole-cell current-clamp recording configuration. Patch electrodes were pulled (micropipette puller P-97, Sutter Instrument, USA) from borosilicate glass (O.D. 1.5 mm, I.D. 0.86 mm, Sutter Instrument) to a resistance of 2–4 mΩ. The pipette internal solution contained [in mM: 125 potassium gluconate, 5 KCl, 10 Hepes, 0.6 EGTA, 2 MgCl$_2$, 7 Phosphocreatine, 3 adenosine-5′-triphosphate (magnesium salt), 0.3 guanosine-5′-triphosphate (sodium salt) (pH adjusted to 7.25 with KOH; osmolarity 300 mOsm/l adjusted with d-Mannitol)] and added with biocytin 0.4% (liquid junction potential −14.8 mV was corrected on the data and statistics).

CNO (10 μM in aCSF) was fast perfused close to the recording cell for 45 s then immediately washed out. Electrophysiological data were recorded using a Multiclamp 700B amplifier (Molecular Devices, UK), low-pass filtered at 4 kHz and digitized at 10 Hz (current clamp) or 4 Hz (voltage clamp) (Digidata 1440A, Molecular Devices, UK). Signals were analyzed offline (Clampfit software, pClamp 10, Molecular Devices, UK). For statistical analysis, "Vehicle" data were collected during the last 60 s before CNO perfusion, then "CNO" data were collected after 45 s of CNO treatment.

**Statistical analysis.** The data (mean ± SEM) were analyzed using the Student *t*-test (two-tailed) and one, two, or three ways ANOVA which was followed by Tukey's comparison test when necessary. Data were tested for normality and Wilcoxon matched-pairs tests and Mann–Whitney test were also used when required. All analyses were carried out using the software GraphPad Prisms 6 and 8.

**Reporting summary.** Further information on research design is available in the Nature Research Reporting Summary linked to this article.

## Data availability

All data supporting the findings of this study are provided within the paper and its supplementary information. A source data file is provided with this paper. The CAG-Gi-IRES-GFP retroviral construct is available upon request to the authors after MTA approval. All additional information will be made available upon reasonable request to the authors.

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

## Acknowledgements
In memory of our colleague and friend, Federico Massa. The authors thank Dr. F. Massa and Dr. G. Marsicano for lending their electrophysiology equipment. We greatly acknowledge C. Dupuy for animal care. Supported by Inserm (to D.N.A.), ANR (to S.T. ANR-16-CE37-0018-01; to D.N.A. ANR-Blanc-1408-01; to D.C. ANR-13-BSV4-0006-01)—M.L. was supported by a MESR (Ministère de l'Enseignement Supérieur et de la Recherche) fellowship and by the ANR (ANR-16-CE37-0018-01). N.M. was supported by the ANR (ANR-16-CE37-0018-01) and by the Fondation Fyssen. We thank Dr. Fred Gage and Dr. Dieter Chichung Lie for providing the retroviral vector CAG–GFP and the 293GPG cell line, respectively. This work benefited from the support of the Biochemistry and Biophysics Facility of the Bordeaux Neurocampus funded by the LabEX BRAIN ANR-10-LABX-43 and the Animal Housing facility funded by Inserm and LabEX BRAIN ANR-10-LABX-43. The confocal analysis was done in the Bordeaux Imaging Center (BIC), a service unit of the CNRS-INSERM and Bordeaux University, member of the national infrastructure France BioImaging supported by the French National Research Agency (ANR-10-INBS-04).

## Author contributions
M.L. designed and performed the experiments and analyzed the data. E.P. designed the retroviruses. F.F. produced the retroviruses. W.M. and V.C. performed the electrophysiological experiments. D.C. and F.M. supervised electrophysiological experiments. P.M. and N.M. performed the experiments. G.F. designed the experiments and revised the paper. D.N.A. conceived experiments and wrote the paper. S.T. performed, conceived, and designed the experiments, analyzed the data and wrote the paper. All the authors edited and approved the final version of the manuscript.

## Competing interests
The authors declare no competing interests.
