## [Peer Review File · Nature Communications]

Reviewers' Comments:

Reviewer #1:

Remarks to the Author:

In this manuscript, the authors investigate how the maturity of neuron in the dentate gyrus during learning affects its later involvement in reconsolidation. While both mature and immature neurons (at the time of learning) are activated by a remote memory, the authors report that inactivating neurons that were immature during learning interferes with reconsolidation (inactivating neurons that were mature during learning did not show this). The results are very interesting, however the paper as a whole would greatly benefit from a validation of the methods used. Some examples of these:

1. Validation of the ability to label immature cells with thymidine analogues (and the percent labelled). If this varies between animals, this may affect the results in Fig 1,2.
2. Validation of Zif268 expression (with thymidine labelled cells).
3. Perhaps most importantly the spread, efficacy, and specificity of hM4Di. If the retrovirus is selective for immature cells, it is worth demonstrating this here (even if it has been demonstrated previously). Is all the dentate gyrus transfected? What percentage of immature cells are affected.

Major issues

1. Ideally the authors demonstrate that the neurons affected during reactivation are also involved in learning. But at the very least, the authors need to demonstrate whether mature and/or immature neurons are involved in the initial learning. Ultimately are the results observed during reconsolidation related to the neuron's immaturity or its involvement during learning.
2. Why doesn't CNO (in both Fig 3 and 4) during reconsolidation cause a drop in performance? This is essentially a recall experiment with a portion of the hippocampus inactive.
3. I don't feel the "good performers only" criteria are justified. It is arbitrary where one sets these criteria, and especially problematic if determined post-hoc. It would be more compelling to plot the raw data of prior performance and the effect of CNO on the first test, and measure a correlation.
4. It is unclear why the first test does not support the hypothesis, but a later test does. Would simply having one test with a longer delay show the same effect?
5. Is it possible that DREADDs is not expressed at the same level after 10 weeks (Fig.4). The authors should show a positive control, (or at least a comparison of expression with mice on shorter protocols) given that no effect was observed.
6. I am not sure what Fig 5 and 6 add, as the rats are already going to the platform location from memory (regardless if that memory gets updated or not).

Minor issues

1. Remove the accent on "reactivation" in Fig. 3
2. Fig 3,4 5,6- the CNO image looks like the infusion is taking 2 days...
3. How are the time points determined- for example in Fig 2, the 6 weeks between IdU and learning.
4. Consistency- Black dot for learning maze in Fig 5, and red dot in learning maze in Fig 6

5. Terms such as BrdU should be explained in text.
6. is consolidation (4 weeks post learning) necessary for reconsolidation? In the Carrion and colleagues paper only a few days passed between learning and reconsolidation?
7. Fig 2 (should have a subplot like Fig 1b). Is there a reason not to show the data?
8. young granule cells have more activity and are less spatially tuned. This should be discussed relative to your results.
9. there are a number of labels or explanations missing throughout. An example of missing labels in plots- what "T" and "O" mean in Fig 5 and 6 are only described in the methods section. This should definitely be in the figure legend.

Reviewer #2:

Remarks to the Author:

Lods and colleagues examined the activity patterns and necessity of mature and immature adult-born neurons of the dentate gyrus (DG) to the reconsolidation processes of remote spatial memory. To achieve this, the authors trained male rats in the Morris water maze (MWM), subsequently showing that a probe trial was sufficient to induce protein-synthesis-dependent updating of learned performance. Interestingly, immediate early gene expression (Zif268, in this case) in adult-born cells (labeled with BrdU) was found to be increased in animals undergoing the update, but this was lost in animals treated with anisomycin after the update. The authors show that this increased activity is specific to adult-born cells, but not necessarily to those labeled shortly after birth (p7). Moreover, by using a retroviral approach in which inhibitory DREADDs could be delivered to adult-born cells, the authors show that broad inhibition of immature (1 week old prior to learning), but not mature (6 weeks old), DG cells somewhat recapitulates the effects of anisomycin treatment after reactivation.

Overall, the findings are interesting but are quite phenomenological, we think. The data does not support the main conclusions for a role for new neurons in memory reconsolidation.

Mejor

-Fig.1 shows an effect of Anisomycin on spatial memory performance at T. Fig 2 shows activation patterns at T. But in Fig. 3 silencing of immature neurons (1 week at time of learning) does not affect spatial memory performance at T (unless the authors classify good performers which was NOT done in Fig. 1 and 2).

-What is effect of silencing on activation pattern of new (1 week and 6 week at time of learning) neurons (GFP+ zif268+) during T?

-How does silencing new neurons during R affect zif268 in whole DG or CA3 at T?

- Are 6 week neurons labeled in adult rats not activated to same extent as 1 week neurons labeled in adult rats? Are 6 weeks neurons considered mature?

-Fig. 4b Negative results are hard to interpret. But here, it appears that silencing 6 week neurons increases latency at R (as shown in other studies - silencing of new neurons at time of retrieval impedes spatial memory performance Yan Gu et al Nat Neuro 2012 or Arruda-Carvalho and Frankland JNeuro 2011).

-The conclusions from Fig. 5 and 6 are not supported by data and appear to be driven by unusual behavior of control groups.

Fig. 5b vs Fig. 6b: In Fig 5b the controls do not show preference for T over O in R but in Fig 6, the controls do show preference for T over O in R.

Also, silencing 1 week neurons (at time of learning) appears to increase preference for T over O in R (Fig 5b)?

Minor

- Please include image(s) of BrdU/IdU/CldU/Zif268 labeling, as only images of RV expression are included.
- I appreciate the zone data included in Figures 5b-c and 6b-c, but could the authors also show the latency to first cross the platform?
- It's fine if the authors would rather include these data in the supplementary, but the authors should show the behavioral data (latency/etc.) to the platform for the data in Figure 2 (rather than exclude).
- For the experiments using cannula placements into the lateral ventricles, there is no mention of how proper placement was confirmed; where any animals excluded based on misses? Can the authors elaborate? Similar things could be said about RV injections into the DG; were any animals excluded for their induction/cell count totals? Did cell count totals for induction correlate at all with behavioral performance in the Gi-GFP animals?
- Cite Garthe 2009, Kitamura 2009, Sahay 2011, Snyder 2005, Anacker and Hen 2017

Reviewer #3:

Remarks to the Author:

The manuscript by Lods et al. report on a series of interesting experiments investigating the role of neurogenesis on remote spatial memory reconsolidation. The authors used the rodent Morris water maze task, which is known to heavily engage the hippocampus. After 4 weeks, the reconsolidation of the remote memory was triggered by a reminder session and post-reactivation anisomycin was used to block reconsolidation. Inducing the reconsolidation of remote memories is not always easy. Here, the reactivation session had a mismatch (the platform was removed), which can promote memory destabilization. A test was conducted 2 days later to evaluate the efficacy of the reconsolidation-blockade procedure and the data shows that it was effective in causing amnesia. Then the authors looked at the role of adult-born neurons during remote memory reconsolidation. These neurons were labeled with a BrdU injection 1-week prior training, and then Zif268 expression in labeled cells was quantified after recall. Zif268 is a good marker of activity-induced neuronal plasticity, and thus a good proxy for visualizing neurons engaged by recall and reconsolidation. The results show that adult-born neurons are indeed recruited by recall (i.e. BrdU+ cells expressed zif268). Interestingly, this was completely abolished if reconsolidation was blocked with anisomycin suggesting that these neurons are not only engaged by retrieval but are potentially necessary for re-storage. Adult born neurons that were already mature at the time of training were also recruited by remote memory recall, but this was not the case of neurons generated during early postnatal developmental stages. Appropriate controls showed that learning increases the survival of immature neurons, and overall DG activation by reactivation does not change between groups. Importantly, this effect was not dependent on the animal's age during learning since older rats displayed the same outcome.

To causally assess the involvement of adult-born neurons with reconsolidation the author used a DREADD approach where adult-born neurons generated during learning were silenced during reactivation. This procedure didn't result in amnesia in the first test, although this may be due to data variability since animals displaying good performance during reactivation did develop amnesia. A second test one week later made evident that the DREADD silencing impaired reconsolidation resulting in a weaker memory that didn't persist over time. Interestingly, silencing adult-born neurons that were already mature at the time of learning didn't disrupt reconsolidation. This indicates that when a remote memory becomes labile upon recall, its re-storage depends on the neurons that were generated at the time of initial learning, but not on those that were already mature.

Next, the authors investigated reconsolidation-mediated memory strengthening. Silencing adult-born neurons generated at the time of initial learning prevented memory to be enhanced by reactivation. However, this was not the case when adult-born neurons that were already mature during learning were silenced. This further corroborates with the authors' hypothesis that adult-born neurons generated at the time of memory formation are integrated into the memory trace

and are necessary for reconsolidation at remote time-points.

This manuscript addresses an important topic and provides novel, relevant knowledge about the biology of remote memories. The study is very well designed, controls are impeccable, and the results are convincing. The discussion of the data and of the previous literature is informative and clever. Overall, I'm very positive about this manuscript. I just have a few considerations

- The authors revealed critical mechanisms for spatial memory formation and remote memory reconsolidation involving adult-born neurons. One question that remains to be addressed is whether these mechanisms are specific to remote memories or if it applies to recent memories as well. An immuno at an earlier time-point would help to answer this question. In this regard, a recent paper by Suarez-Pereira and Carrion (2015) suggested that immature adult-born neurons are engaged in the reconsolidation of a recent object recognition memory. This article is very relevant to the current study and should be discussed.

- The literature on reconsolidated-mediated updating and strengthening, and on the role of a mismatch for reconsolidation induction is central to this manuscript. I encourage the authors to briefly summarize this literature and acknowledge papers that helped to build our current understanding of these processes. Critical papers include the first demonstration by Morris et al 2006 that spatial memories become resistant to undergo reconsolidation when no new learning occurs during recall, M Kindt and ME Pedreira works on prediction error, and the reconsolidation-update literature

We would like to thank the referees for the positive comments and constructive criticisms on the manuscript. We have largely reorganized the manuscript and performed experiments in order to address all referees' comments. You will also find below a detailed description of the changes made and comments on certain points raised by the referees.

Reviewer #1 (Remarks to the Author):

In this manuscript, the authors investigate how the maturity of neuron in the dentate gyrus during learning affects its later involvement in reconsolidation. While both mature and immature neurons (at the time of learning) are activated by a remote memory, the authors report that inactivating neurons that were immature during learning interferes with reconsolidation (inactivating neurons that were mature during learning did not show this). The results are very interesting, however the paper as a whole would greatly benefit from a validation of the methods used. Some examples of these:

1. Validation of the ability to label immature cells with thymidine analogues (and the percent labelled). If this varies between animal, this may affect the results in Fig 1,2.
2. Validation of Zif268 expression (with thymidine labelled cells).
3. Perhaps most importantly the spread, efficacy, and specificity of hM4Di. If the retrovirus is selective for immature cells, it is worth demonstrating this here (even if its been demonstrated previously). Is all the dentate gyrus transfected? What percentage of immature cells are affected.

We thank the reviewer for the positive assessment of our results. As the reviewer requested, we added some information for all methods used.

1. Concerning the ability to label immature cells with thymidine analogues, we added some references of papers that validate this technique (Gould et al, 1999; Kempermann et al, 1997; Wojtowicz et al, 2006, Kuhn and Cooper-Kuhn 2007) (p.5). We also added two figures in the supplementary information (Fig S1b and FigS3e) demonstrating that the number of labelled cells does not vary between animals and that this number is not statistically different between the groups in Fig 1 and S3. In other words, the aCSF-R, Ani-R and Ani groups have an equivalent number of BrdU or IdU positive cells.

2. Validation of Zif268 expression (with thymidine labelled cells).

We added in the manuscript all the references of papers by our group and by others that have used this functional imaging approach to study new neuron activation. In particular this approach has been validated in a Nature protocol paper (Kee et al, Nat protocol, 2007) (p.5).

3. Perhaps most importantly the spread, efficacy, and specificity of hM4Di. If the retrovirus is selective for immature cells, it is worth demonstrating this here (even if its been demonstrated previously). Is all the dentate gyrus transfected? What percentage of immature cells are affected.

In order to validate that the retrovirus hM4Di did not transduce mature cells, we performed a double-staining for GFP and doublecortin (DCX). DCX is a specific marker for immature

neurons as it is only expressed by post-mitotic neurons and up to 2-3 weeks after cell birth. As the results show, more than 98% of the GFP cells express DCX demonstrating that our DREADDs-retrovirus is specific for immature neurons (**Fig 3g**). To address the percentage of immature cells affected we performed another experiment in which rats were injected in the DG with the retroviruses (Gi-GFP and GFP) and with BrdU (ip). Because the integration kinetics of both markers are very different, we could not analyse the percentage of colocalization between BrdU and GFP. However, by counting the number of BrdU-IR cells and GFP-IR cells we can assume that more than 60% of the population of cells that were tagged one week before learning by BrdU are GFP positive (number of GFP-IR cells/number of BrdU-IR cells X 100). This percentage is quite important considering that mostly the septal dentate gyrus is infected. Concerning this point, we added a figure in supplementary information (**Fig S8**) showing the spread of the infection. As shown in this figure, the septal dentate gyrus is infected and the GFP labelled cells extent from -2mm to -5mm/6mm from Bregma.

Major issues

1. Ideally the authors demonstrate that the neurons affected during reactivation are also involved in learning. But at the very least, the authors need to demonstrate whether mature and/or immature neurons are involved in the initial learning. Ultimately are the results observed during reconsolidation related to the neuron's immaturity or its involvement during learning.

By using ablation technics (such as transgenesis or pharmacological depletion) it has been shown that adult-born neurons that are mature at the time of training are involved in spatial learning which it is not the case of immature neurons (Lemaire et al, 2012; Dupret et al 2008). However different results have been obtained when manipulating the activity of the neurons. Indeed, reversible silencing of adult-born neurons during learning does not affect memory acquisition but memory expression. In fact, as discussed by Gu et al, (Nature Neurosciences , 2012) *“spatial learning can occur in the absence of newborn neurons, but if newborn neurons are present and functional in circuit level at the time of training, they are recruited into hippocampal memory circuits and silencing (or ablating) these cells revealed that they are essential for memory retrieval”*.

So, to address the issue raised by the reviewer, we performed two experiments (**Fig 3**). To determine whether immature adult-born neurons were involved in the initial learning, BrdU and the retroviruses (GFP and Gi-GFP) were injected in rats one week before learning. Rats were trained in the WM for a week and tested 48h later. They received CNO 1 hour before the test to silence the immature neurons infected. The results showed that silencing this population during the test had no impact on memory retrieval and that this population is not activated by the test, demonstrating that immature neurons were not recruited by the initial learning (**Fig 3c-f**). Then, we injected the retroviruses 6 weeks before learning, trained the rats in the WM for a week and tested them 48h later under CNO. The results showed that inhibiting this mature population affects memory retrieval demonstrating that they are actually involved in the initial learning (**Fig 3a,b**). In conclusion, the results observed during reconsolidation are related to the neurons' immaturity during learning.

2. Why doesn't CNO (in both Fig 3 and 4) during reconsolidation cause a drop in performance? This is essentially a recall experiment with a portion of the hippocampus inactive.

According to the literature, the CNO kinetic of action is unclear when CNO is injected i.p. However, a study from Garner et al (Science 2012) demonstrated that neuronal activity of DREADDs-transduced cells was optimally affected 30-40 minutes after i.p. injection. We therefore chose to inject CNO 30 min before the test in order to inhibit the transduced cells right after the reactivation session. Our first results (**described in Fig 4**) demonstrated that CNO does not impact the performances at the reactivation test. This suggests that the transduced cells are not inactivated yet. In order to prove that our chemogenetic manipulation can affect retrieval, we performed a new experiment in which mature neurons are transduced with the DREADD-retrovirus and in which rats were trained in the WM and CNO was injected 1h before a probe test. The results showed that memory is impaired when CNO is injected 1h before the test (**Fig 3a,b**). This suggests that it takes between 30min and 1hour for the CNO to be effective.

3. I don't feel the "good performers only" criteria are justified. It is arbitrary where one sets these criteria, and especially problematic if determined post-hoc. It would be more compelling to plot the raw data of prior performance and the effect of CNO on the first test, and measure a correlation.

Since we evaluated reconsolidation of remote memory, the probe test (reactivation) was performed 4 weeks after WM training. After such a long delay, few animals (15-20%) did not retain the precise location of the platform and required more than 30 sec to first cross this position. Interestingly, a study by Ramirez-Amaya et al, 2001 indicates that memory performance 30 days after WM training is only correlated with hippocampal synaptogenesis, i.e. a cellular marker of spatial representation storage, when the latency to first cross the virtual platform location is below 30 sec. As reconsolidation can only occur if memory has been well consolidated (Eisenberg and Dudai, 2003) we decided to exclude the few animals using the criterion of 30 sec to only select "good performers". Furthermore, we added a paragraph to justify this 30sec criteria (**p4**).

This criteria was used in the first experiments when we analyzed the Zif268 expression after anisomycin treatment (**Fig 1,2**). Since anisomycin was injected right after the reactivation test, the performances during reactivation were not influenced by the injection. In the second set of experiments (when using the DREADDs-retrovirus) CNO injection was performed before the reactivation. Importantly, CNO had no impact on memory retrieval allowing us to apply the same criteria (**Fig 4b**). Furthermore, our supplementary experiment confirmed that CNO is not effective on retrieval when injected 30min before the test (**Fig 3b**).

4. It is unclear why the first test does not support the hypothesis, but a later test does. Would simply having one test with a longer delay show the same effect?

We performed a first test (2 days after reactivation) to be exactly in the same conditions than the ones in the first activation experiments. Since the results of the first test revealed a slight impairment we decided to perform a second test 2 weeks later. This second test was done to demonstrate that the impairment was long lasting. Furthermore, it shows that reconsolidation (such as consolidation) is a time-dependent process. But the reviewer is right, a first test with a longer delay should show the same effect i.e a long-term memory impairment. In addition we performed a new experiment and using the 30 sec criterion we replicated our chemogenetic effect on the first test (**Fig S9b**).

5. Is it possible that DREADDs is not expressed at the same level after 10 weeks (Fig.4). The authors should show a positive control, (or at least a comparison of expression with mice on shorter protocols) given that no effect observed.

As mentioned in the text, the number of cells transduced is 1469 (**Fig 4**) and 1871 (**Fig 6**) for neurons that are immature at the time of learning (short delay experiments) and 1934 (**Fig 5**) and 1678 (**Fig 7**) for neurons that are mature at the time of learning (long delay experiments). To reinforce this point we added a figure (**Fig S10**) in the supplementary information to show the number of transduced cells. As the figure shows there is no difference between the number of transduced cells and their age after transduction.

6. I am not sure what fig 5 and 6 add, as the rats are already going to the platform location from memory (regardless if that memory gets updated or not).

To address this point, we added more explanation in the text. In the classical protocol used in Fig 3,4 and 5, at the reactivation trial, rats went directly to the position where the platform was during training. When we used the Atlantis protocol we can see that rats went also to the platform position but they did not persist in searching in the target position due to a loss of accuracy induced by the 4 weeks delay. However, we can observe that memory is reinforced or updated during the “Atlantis” reactivation. This reactivation leads to an increase in the accuracy of memory that can be observed in the increase number of entrance and time spent in the target zone. This protocol was used to show that immature neurons are involved in both hallmarks of reconsolidation : memory maintenance (with a non-reinforced protocol) and update of memory (with a reinforced protocol). (**P14, 1st paragraph**)

Minor issues

1. Remove the accent on “reactivation” in Fig. 3

We corrected this mistake

2. Fig 3,4 5,6- the CNO image looks like the infusion is taking 2 days...

We modified the figures to avoid any confusion .

3. How are the time points determined- for example in fig 2, the 6 weeks between IdU and learning.

We have previously shown that adult-born neurons are mature and activated by learning at the age of 6 weeks (Tronel et al, Hippocampus 2015). We could have chosen a longer delay (2 months or more) but it would have increased even more the time length of the experiment. Concerning the 1 week delay, we focused on the age of immature neurons because we have shown that the survival of this specific population is increased by spatial learning and that learning regulates the development and the maturation of these neurons (Dupret et al, PLoS Bio 2007, Tronel et al, PNAS 2010). We added a sentence in the text to justify the time points used. (p6).

4. Consistency- Black dot for learning maze in fig 5, and red dot in learning maze in fig 6

We apologize for the mistake, we corrected it.

5. Terms such as BrdU should be explained in text.

We added the explanation.

6. is consolidation (4 weeks post learning) necessary for reconsolidation? In the Carrion and colleagues paper only a few days passed between learning and reconsolidation?

Reconsolidation is a process that happens when a consolidation memory is reactivated and thus returns to a labile state. Therefore, only a consolidated memory can be reconsolidated. Consolidation occurs through time. In both water maze (our results on recent reconsolidation experiments, **Fig 3a-d**) and object recognition (Suarez-Pereira and Carrion, 2015) memory is consolidated two days after training. As discussed in our paper, many studies have shown that recent spatial memory undergoes reconsolidation but this is the first evidence that remote memory can also undergo reconsolidation.

7. Fig 2 (should have a subplot like Fig 1b). Is there a reason not to show the data?

We added the figure in Fig S2a.

8. Young granule cells have more activity and are less spatially tuned. This should be discussed relative to your results.

We completed the discussion to address this point. In particular, we discussed our results in lights of a study published by Danielson et al in Neuron in 2016 and a review of Aimone et al, 2014 on the role of immature neurons (p19-20).

9. There are a number of labels or explanations missing throughout. An example of missing

labels in plots- what “T” and “O” mean in Fig 5 and 6 are only described in the methods section. This should definitely be in the figure legend.

We added the description of the labels in the figure legends.

Reviewer #2 (Remarks to the Author):

Lods and colleagues examined the activity patterns and necessity of mature and immature adult-born neurons of the dentate gyrus (DG) to the reconsolidation processes of remote spatial memory. To achieve this, the authors trained male rats in the Morris water maze (MWM), subsequently showing that a probe trial was sufficient to induce protein-synthesis-dependent updating of learned performance. Interestingly, immediate early gene expression (Zif268, in this case) in adult-born cells (labeled with BrdU) was found to be increased in animals undergoing the update, but this was lost in animals treated with anisomycin after the update. The authors show that this increased activity is specific to adult-born cells, but not necessarily to those labeled shortly after birth (p7). Moreover, by using a retroviral approach in which inhibitory DREADDs could be delivered to adult-born cells, the authors show that broad inhibition of immature (1 week old prior to learning), but not mature (6 weeks old), DG cells somewhat recapitulates the effects of anisomycin treatment after reactivation.

Overall, the findings are interesting but are quite phenomenological, we think. The data does not support the main conclusions for a role for new neurons in memory reconsolidation.

Mejor

-Fig.1 shows an effect of Anisomycin on spatial memory performance at T. Fig 2 shows activation patterns at T. But in Fig. 3 silencing of immature neurons (1 week at time of learning) does not affect spatial memory performance at T (unless the authors classify good performers which was NOT done in Fig. 1 and 2).

We rephrased our result sections. In the first experiment, only good performers were treated with anisomycin as it was mentioned in the method section. Reconsolidation can only occur if memory has been consolidated in the first place. Therefore, only rats that reach the position of where the platform was received anisomycin injection. For the DREADD-inhibition experiments CNO was injected before the reactivation trial so all rats were treated. To be able to apply the same criteria we had to make sure that CNO had no effect on behavioural performances observed at the reactivation trial (as the results showed on Fig 3). To reinforce this point we performed another experiment in which CNO was injected either 1hour or 30 min before a test. As the results showed (**Fig 3b**), memory is impaired only when CNO was injected 1hour before a test demonstrating that the compound is not effective when injected 30 min before. This allow us to consider that rats that did not cross the position of the platform within the first 30 seconds did not properly learn or retain the task.

-What is effect of silencing on activation pattern of new (1 week and 6 week at time of learning) neurons (GFP+ zif268+) during T?

We thank the reviewer for this excellent question. To address it we replicated our experiment in which immature neurons labelled with the Gi-GFP-RV were inhibited. As expected, we replicated our results since memory was impaired during the test. Since rats were injected with BrdU, we analysed the expression of BrdU-Zif268 when rats are killed 90 min after the test. The results showed that the activation is decreased in the Gi-GFP-RV rats compared to that observed in GFP-RV rats (**Fig S9**). Concerning the activation of mature adult-born neurons, we added another experiment in which this population was silenced during recent memory reconsolidation. The results showed that memory reconsolidation was not affected and this analysis of BrdU-Zif268 expression revealed that mature adult-born neurons' activation was not affected (**Fig S11**).

-How does silencing new neurons during R affect zif268 in whole DG or CA3 at T?

We did not analyse Zif268 expression in the whole DG or CA3 as it is a time consuming procedure and we were running out of time because of the COVID-19 lockdown, but we have previously published that a decrease of activation of adult-born neurons does not induce a decrease of activation in the whole DG (Tronel et al, Hippocampus 2015 and Tronel et al, Brain Struct Funct, 2015). This underlines that the decrease of activation is specific of adult-born neurons and that the behavioural effects do not result from a global effect of the circuit.

- Are 6 week neurons labeled in adult rats not activated to same extent as 1 week neurons labeled in adult rats? Are 6 weeks neurons considered mature?

To address the comment on the activation of 1-week-old neurons we added an experiment in which immature neurons were injected with BrdU one week before learning. Rats were trained in the WM for one week and memory was tested 2 days later. Animals were sacrificed 90 min after the test. The analyse of Zif268 expression in BrdU-IR cells demonstrated that this population was not activated.

Answering the question of whether 6 week-old neurons are mature is challenging. In rats new neurons mature faster compared to those in mice (Snyder et al, 2009). At 6 weeks adult-born neurons are functionally integrated into the DG network and are recruited by learning (Tronel et al, 2015). However, they continue to develop for at least 6 months (Cole et al, 2020), which may explain why they are still plastic even if considered mature (Lemaire et al, 2012). Therefore we consider them as mature.

-Fig. 4b Negative results are hard to interpret. But here, it appears that silencing 6 week neurons increases latency at R (as shown in other studies - silencing of new neurons at time of retrieval impedes spatial memory performance Yan Gu et al Nat Neuro 2012 or Arruda-Carvalho and Frankland JNeuro 2011).

We performed a new experiment to evaluate the role of mature neurons in recent spatial memory retrieval as was previously done by Gu et al, 2012 and Arruda-carvalho and Frankland, 2011. CNO injection 1 h before the probe test impaired memory performances (Fig 3b), a result in agreement with the findings in the others studies. However, CNO injection 30 min before the test had no impact on memory retention.

Although chemogenetic silencing of mature neurons seems to increase latency at remote reactivation (Fig 5b), it is not statistically significant.

We performed a two ways Anova with repeated measures on the performances between day 6 and the reactivation trial, the results showed no significant effect on either groups ($p=0.6935$), time ($p=0.3081$) or interaction ($p=0.6741$). In addition when we performed an unpaired t-Test between the groups at the reactivation trial, the results showed no difference ($p=0.3295$).

-The conclusions from Fig. 5 and 6 are not supported by data and appear to be driven by unusual behavior of control groups. Fig. 5b vs Fig. 6b: In Fig 5b the controls do not show preference for T over O in R but in Fig 6, the controls do show preference for T over O in R. Also, silencing 1 week neurons (at time of learning) appears to increase preference for T over O in R (Fig 5b)?

Once again, we understand the comment of the reviewer. In Fig 6 (Fig 5 in previous version of the manuscript) the reviewer suggests that silencing immature neurons increases preference for T over O. However the statistical analysis demonstrated that there is no difference between the Gi rats and the GFP rats in the number of annulus crossing or the time spent in zones at the reactivation trial. The conclusions from this figure is that the Atlantis reactivation induces an increase of performances for GFP rats between reactivation and test. This is supported by the statically difference between performances for the T zone (in grey) between R and T ($***p<0.001$). In contrast we do not see this increase in Gi rats (performances for the T zone, (in red), are not different between R and T).

Concerning the second comment, it is true that when we performed the experiment with the rats in which the mature population was targeted by the retrovirus, the results were different. At the reactivation trial, both groups display a preference for the target zone. However, the statistical analysis showed that both groups had an increase of performances at the test induced by the "Atlantis" reactivation. Therefore, we think that our data support our conclusion that is the inhibition of mature neurons has no impact on the memory update induced by reconsolidation.

Minor

- Please include image(s) of BrdU/IdU/CldU/Zif268 labeling, as only images of RV expression are included.

We added some confocal illustration of BrdU/IdU/CldU/Zif268 and DCX/GFP labelling in Fig 1,2,3.

- I appreciate the zone data included in Figures 5b-c and 6b-c, but could the authors also show the latency to first cross the platform?

We added the figure showing the latency to cross the platform in Fig S12.

- It's fine if the authors would rather include these data in the supplementary, but the authors should show the behavioral data (latency/etc.) to the platform for the data in Figure 2 (rather than exclude).

We added this figure in the supplementary information Fig S2.

- For the experiments using cannula placements into the lateral ventricles, there is no mention of how proper placement was confirmed; where any animals excluded based on misses? Can the authors elaborate? Similar things could be said about RV injections into the DG; were any animals excluded for their induction/cell count totals? Did cell count totals for induction correlate at all with behavioral performance in the Gi-GFP animals?

We added this information in the Materials and Methods sections. Cannula implantation was checked on the brain slices and animals were removed for further analysis when cannula was incorrectly positioned. Concerning the retroviruses injections, animals were removed from the experiment when GFP-labelled cells were not or unilaterally observed. **(Mat & Mat , Surgery)**

We never observed any correlation between the number of GFP labelled cells and behavioral performances.

- Cite Garthe 2009, Kitamura 2009, Sahay 2011, Snyder 2005, Anacker and Hen 2017

We added these references in our manuscript.

Reviewer #3 (Remarks to the Author):

The manuscript by Lods et al. report on a series of interesting experiments investigating the role of neurogenesis on remote spatial memory reconsolidation. The authors used the rodent Morris water maze task, which is known to heavily engage the hippocampus. After 4 weeks, the reconsolidation of the remote memory was triggered by a reminder session and post-reactivation anisomycin was used to block reconsolidation. Inducing the reconsolidation of remote memories is not always easy. Here, the reactivation session had a mismatch (the platform was removed), which can promote memory destabilization. A test was conducted 2 days later to evaluate the efficacy of the reconsolidation-blockade procedure and the data shows that it was effective in causing amnesia.

Then the authors looked at the role of adult-born neurons during remote memory reconsolidation. These neurons were labeled with a BrdU injection 1-week prior training, and then Zif268 expression in labeled cells was quantified after recall. Zif268 is a good marker of activity-induced neuronal plasticity, and thus a good proxy for visualizing neurons engaged by recall and reconsolidation. The results show that adult-born neurons are indeed recruited by recall (i.e. BrdU+ cells expressed zif268). Interestingly, this was completely abolished if reconsolidation was blocked with anisomycin suggesting that these neurons are

not only engaged by retrieval but are potentially necessary for re-storage. Adult born neurons that were already mature at the time of training were also recruited by remote memory recall, but this was not the case of neurons generated during early postnatal developmental stages. Appropriate controls showed that learning increases the survival of immature neurons, and overall DG activation by reactivation does not change between groups. Importantly, this effect was not dependent on the animal's age during learning since older rats displayed the same outcome.

To causally assess the involvement of adult-born neurons with reconsolidation the author used a DREADD approach where adult-born neurons generated during learning were silenced during reactivation. This procedure didn't result in amnesia in the first test, although this may be due to data variability since animals displaying good performance during reactivation did develop amnesia. A second test one week later made evident that the DREADD silencing impaired reconsolidation resulting in a weaker memory that didn't persist over time. Interestingly, silencing adult-born neurons that were already mature at the time of learning didn't disrupt reconsolidation. This indicates that when a remote memory becomes labile upon recall, its re-storage depends on the neurons that were generated at the time of initial learning, but not on those that were already mature. Next, the authors investigated reconsolidation-mediated memory strengthening. Silencing adult-born neurons generated at the time of initial learning prevented memory to be enhanced by reactivation. However, this was not the case when adult-born neurons that were already mature during learning were silenced. This further corroborates with the authors' hypothesis that adult-born neurons generated at the time of memory formation are integrated into the memory trace and are necessary for reconsolidation at remote time-points.

This manuscript addresses an important topic and provides novel, relevant knowledge about the biology of remote memories. The study is very well designed, controls are impeccable, and the results are convincing. The discussion of the data and of the previous literature is informative and clever. Overall, I'm very positive about this manuscript. I just have a few considerations

- The authors revealed critical mechanisms for spatial memory formation and remote memory reconsolidation involving adult-born neurons. One question that remains to be addressed is whether these mechanisms are specific to remote memories or if it applies to recent memories as well. An immuno at an earlier time-point would help to answer this question. In this regard, a recent paper by Suarez-Pereira and Carrion (2015) suggested that immature adult-born neurons are engaged in the reconsolidation of a recent object recognition memory. This article is very relevant to the current study and should be discussed.

We thank the reviewer for his/her encouraging comments on our study. To address the question on the specificity of the mechanisms to remote memories, we performed two supplementary experiments. In the first experiment, we showed that immature neurons at the time of training are not activated during recent memory retrieval (Fig 3e), contrary to what we found for remote memory retrieval (Fig 1d). These results demonstrate that the role of immature neurons is specific to remote memory reconsolidation.

In a second experiment, we targeted and silenced mature neurons during recent reconsolidation (memory was reactivated 2 days after training). We found no effect on recent memory reconsolidation (Fig S11). This result indicates that mature adult-born neurons are not involved in recent spatial memory reconsolidation. Furthermore, we added a paragraph in the discussion in which we discuss the results of Suarez-Perreira and Carrion **(p18)**.

- The literature on reconsolidated-mediated updating and strengthening, and on the role of a mismatch for reconsolidation induction is central to this manuscript. I encourage the authors to briefly summarize this literature and acknowledge papers that helped to build our current understanding of these processes. Critical papers include the first demonstration by Morris et al 2006 that spatial memories become resistant to undergo reconsolidation when no new learning occurs during recall, M Kindt and ME Pedreira works on prediction error, and the reconsolidation-update literature

We added a paragraph on this literature in the discussion of our manuscript and mentioned the work of M. Kindt and ME Pedreira on prediction errors **(p16)**.

Reviewers' Comments:

Reviewer #1:

Remarks to the Author:

The revised manuscript is much improved, and benefits greatly from the multiple new experiments the authors have added. I am happy with the current version except for one minor point. I understand the justification for the "good performers", but am still a bit uncomfortable with this. Perhaps a middle ground is to keep these figures, but move them to supplementary (so they can still be discussed in the main text). Other than that, great job on the manuscript. It is a considerable amount of work yielding very interesting results.

Reviewer #2:

Remarks to the Author:

The authors have addressed some of the concerns, done a lot of work here but the data simply does not support the main conclusions of paper. The main problem with experimental design and interpretation of results is the Central Assumption "that the population of neurons that are silenced is part of the memory engram associated with the spatial learning task". This is not known, and the authors have not even done the requisite experiments to prove that it may be the case.

The writing of the manuscript (including title, main text, discussion, for example see below) and experimental design seems to suggest that authors interpret Zif268+BrdU labeling as cellular reactivation. This is factually incorrect and is a fundamental flaw in this study. All Zif268+BrdU labeling says is that cells of a certain age are activated during recall. It says nothing about cellular reactivation. This is a critical point as authors make erroneous inferences on reconsolidation because of this flawed interpretation of cellular reactivation. To make claims of participation in reconsolidation, authors have to equate memory reactivation and cellular participate /contribute to reconsolidation (unless it is some general circuit disruption, in which case evidence needs to be shown and the take home message is different).

a. In Figure 4, authors show that Chemogenetic silencing new neurons 1 week at time of learning and 5 weeks at time of memory reactivation R, does not affect recall at R, or 2 days later at (T), but affects recall two weeks later at T2 (memory persistence) (Fig. 4B).

-However, Fig S9b, the authors use the same experimental design and demonstrate impaired retrieval at time point T.

-Also, not clear why the authors have not provided data for RV GFP+Zif268 when they have this data and continue to show only BrdU/Zif268. Please provide this critical data.

b. Fig. 5 The authors claim that chemogenetic silencing new neurons 6 week at time of learning and memory reactivation R does not affect R, T or T2. Sorry, but Looking at Fig. 5b, it does appear that there is an effect on recall during reactivation. Am I missing something here?

c. As a thought experiment, let us agree with authors' claims and consider that immature neurons contribute to memory reconsolidation. How does silencing of 5 week neurons that were too young (1 week old) to participate during learning affect memory persistence (T2) or Recall at T? The more conservative interpretation is that silencing these neurons causes general disruption of circuit, maybe in DG or in target region CA3. Although requested in prior round, the authors have not provided any analysis of Zif268 in DG and in the target, CA3?

d. Fig. 6b & C: The authors say that chemogenetic silencing adult generated neurons 1 week at time of learning and 5 weeks at time of memory reactivation R impairs behavioral updating. How can authors conclude this if statistically speaking, both groups behave in the same way.

Minor:

a. Do we know if dentate neurons or many other dividing cells (astrocytes) are labeled with CldU at P7—Please show images.

b. The authors say: “The fact that CNO does not alter memory expression when injected 30min before test indicates that any effect of such injection on subsequent tests would result from action after reactivation, i.e. during memory reconsolidation”.

This is not true. The lack of an effect may relate to dynamics of CNO rather than a biological mechanism of memory.

c. It is not clear to me what Fig 3 and Fig S7 convey. How can one week old neurons contribute to memory encoding or retrieval, let alone reconsolidation?

Reviewer #3:

Remarks to the Author:

The manuscript entitled "Immature adult-born neurons primed by learning are necessary for remote memory reconsolidation" is the first causal of stem cells involvement in memory reconsolidation. This work is beautifully designed like all of her experimental designs. The current version is acceptable.

Congrats to Prof. Tronel for her first empirical demonstration of stem cell in reconsolidation in remote memory.

This is a grand slam publication for this worthy journal.

REVIEWER COMMENTS

Reviewer #1 (Remarks to the Author):

The revised manuscript is much improved, and benefits greatly from the multiple new experiments the authors have added. I am happy with the current version except for one minor point. I understand the justification for the "good performers", but am still a bit uncomfortable with this. Perhaps a middle ground is to keep these figures, but move them to supplementary (so they can still be discussed in the main text). Other than that, great job on the manuscript. It is a considerable amount of work yielding very interesting results.

We thank the reviewer for its positive comments. As suggested, we moved the figures with the "Good performers" in supplementary information.

Reviewer #2 (Remarks to the Author):

The authors have addressed some of the concerns, done a lot of work here but the data simply does not support the main conclusions of paper. The main problem with experimental design and interpretation of results is the Central Assumption "that the population of neurons that are silenced is part of the memory engram associated with the spatial learning task". This is not known, and the authors have not even done the requisite experiments to prove that it may be the case.

The reviewer is right that immature neurons at the time of learning cannot be considered part of the memory engram based on the seminal definition of "engram cells" (Tonegawa et al., 2015), indicating these cells "are a population of neurons that are activated by learning". We rather suggest that these neurons that are immature at the time of learning are influencing the memory engram network at the time of retrieval when they are mature (6 weeks-old). We therefore rephrased and clarified the corresponding part of the discussion (page 19).

The writing of the manuscript (including title, main text, discussion, for example see below) and experimental design seems to suggest that authors interpret Zif268+BrdU labeling as cellular reactivation. This is factually incorrect and is a fundamental flaw in this study. All Zif268+BrdU labeling says is that cells of a certain age are activated during recall. It says nothing about cellular reactivation. This is a critical point as authors make erroneous inferences on reconsolidation because of this flawed interpretation of cellular reactivation. To make claims of participation in reconsolidation, authors have to equate memory reactivation and cellular participate /contribute to reconsolidation (unless it is some general circuit disruption, in which case evidence needs to be shown and the take home message is different).

We agree with the reviewer that Zif268 expression is a marker of activation (in our case retrieval-induced activation) and not reactivation. We are sorry for the misunderstanding and to further clarify the fact that Zif268 is not a marker of cellular reactivation, we modified the discussion accordingly replacing "reactivated" by "activated" (page 17, 20 in red).

a. In Figure 4, authors show that Chemogenetic silencing new neurons 1 week at time of learning and 5 weeks at time of memory reactivation R, does not affect recall at R, or 2 days later at (T), but affects recall two weeks later at T2 (memory persistence) (Fig. 4B).

-However, Fig S9b, the authors use the same experimental design and demonstrate impaired retrieval at time point T.

-Also, not clear why the authors have not provided data for RV GFP+Zif268 when they have this data and continue to show only BrdU/Zif268. Please provide this critical data.

We clarified this discrepancy in the manuscript. In Fig S9, all rats reached the position of the platform within the 30 first sec. In this case there were considered all as good performers. Therefore, the effect was observed as soon as the 48h test (T) as it was observed in Fig 4d.

For the sake of clarity, we decided to display the previous Fig S9 as main Fig 4, showing first the effect of chemogenetic silencing of neurons that were immature at the time of learning on the 48h-post reactivation test. Then we combined previous Fig 4 and 5, as new Fig 5, to compare mature and immature neurons at the time of learning. In this new Figure 5, chemogenetic silencing of immature neurons at the time of learning does not affect recall at T but (as mentioned in the manuscript p13) in Fig 4, all rats were good performers since they crossed the platform position within the first 30 sec. Therefore, the data from the new fig 4 and the data from the good performers from fig S10 (Fig 4d in the previous version) are similar.

We also added the data showing Zif268 expression in GFP-IR cells. These data are now presented in Fig 4 f,g. The results are similar to those obtained with BrdU-Zif showing that at the post reactivation test, activation of neurons that were immature at the time of learning (GFP-IR cells) is higher compared to those of home cage control and those that were silenced during reconsolidation (Gi-GFP group).

b. Fig. 5 The authors claim that chemogenetic silencing new neurons 6 week at time of learning and memory reactivation R does not affect R, T or T2. Sorry, but Looking at Fig. 5b, it does appear that there is an effect on recall during reactivation. Am I missing something here?

As previously indicated, a two-way Anova with repeated measures on the performances between day 6 and the reactivation trial did not reveal any significant effect on either groups ($p=0.6935$), time ($p=0.3081$) or interaction ($p=0.6741$). In addition, when we performed an unpaired t-Test between the Gi-GFP and GFP groups at the reactivation trial, the results showed no difference ($p=0.3295$). Therefore, although chemogenetic silencing of mature neurons seems to increase latency at remote reactivation (Fig 5b), it is not statistically significant.

Moreover, we performed in the previous version a new experiment to evaluate the role of mature neurons in recent spatial memory retrieval (Fig. 3a,b). CNO injection 1 h (instead of 30min) before the probe test impaired memory performances (Fig 3b) as previously demonstrated by Gu et al, 2012 and Arruda-Carvalho & Frankland, 2011. Again, CNO injection 30 min before the test had no impact on retrieval (Fig 3b).

c. As a thought experiment, let us agree with authors' claims and consider that immature neurons contribute to memory reconsolidation. How does silencing of 5 week neurons that were too young (1 week old) to participate during learning affect memory persistence (T2) or Recall at T? The more conservative interpretation is that silencing these neurons causes general disruption of circuit, maybe in DG or in target region CA3. Although requested in prior round, the authors have not provided any analysis of Zif268 in DG and in the target, CA3?

Chemogenetic silencing of the immature neurons did not induce any effect on global DG or CA3 Zif268 activation, indicating that such manipulation did not induce a general disruption of the DG-CA3 circuit. These results are now provided on FigS9 and discussed page 12.

d. Fig. 6b & C: The authors say that chemogenetic silencing adult generated neurons 1 week at time of learning and 5 weeks at time of memory reactivation R impairs behavioral updating

How can authors conclude this if statistically speaking, both groups behave in the same way.

Looking at the statistical tables, it is clear that comparison of GFP and Gi-GFP groups that were immature at the time of learning show a significant time X zone X group interaction for both annulus crossings ($p=0.0302$) and time in zone ($p=0.0494$). This was not the case for GFP and Gi-GFP groups that were mature at the time of learning with non-significant time X zone X group interaction for both annulus crossings ($p=0.8449$) and time in zone ($p=0.8117$).

For the sake of clarity, we have modified Figures 6 and 7 so that the relevant differences can be easily observed.

Minor:

a. Do we know if dentate neurons or many other dividing cells (astrocytes) are labeled with CldU at P7—Please show images.

We have previously shown that cells labeled at P7 differentiate mainly into neurons (Montaron et al, 2020). However, we added some analysis and image demonstrating that developmentally-generated cells labeled with CldU are neurons for the vast majority (>98%). This is now reported on FigS3.

b. The authors say: "The fact that CNO does not alter memory expression when injected 30min before test indicates that any effect of such injection on subsequent tests would result from action after reactivation, i.e. during memory reconsolidation".

This is not true. The lack of an effect may relate to dynamics of CNO rather than a biological mechanism of memory.

The reviewer is right. We therefore toned down the sentence accordingly: "This rules out that effect of CNO injected 30 min before reactivation on subsequent tests would result from a direct action on retrieval, but more likely after retrieval, i.e. during memory reconsolidation" (page 10).

c. It is not clear to me what Fig 3 and Fig S7 convey. How can one week old neurons contribute to memory encoding or retrieval, let alone reconsolidation?

This experiment was requested by Reviewer 3 on the previous revision step. This shows that

manipulating immature neurons has no effect on recent memory retrieval.

Reviewer #3 (Remarks to the Author):

The manuscript entitled "Immature adult-born neurons primed by learning are necessary for remote memory reconsolidation" is the first causal of stem cells involvement in memory reconsolidation. This work is beautifully designed like all of her experimental designs. The current version is acceptable.

Congrats to Prof. Tronel for her first empirical demonstration of stem cell in reconsolidation in remote memory. This is a grand slam publication for this worthy journal.

Reviewers' Comments:

Reviewer #2:

Remarks to the Author:

1. I am glad that the authors found the comments useful and have performed important controls and edits to convey the distinction between reactivation and activation during retrieval. All of that stated, please change title to convey the main finding. The current title "Immature adult-born neurons primed by learning are necessary for remote memory reconsolidation" is too speculative since there is no evidence that the immature neurons are primed during learning and the authors agree**. This is a discussion point and indeed, the authors discuss this idea.

Instead, the title should reflect the main finding: "Adult-born neurons immature during learning are necessary for remote memory reconsolidation". This is the only objective unbiased way forward.

**Author response: "The reviewer is right that immature neurons at the time of learning cannot be considered part of the memory engram based on the seminal definition of "engram cells" (Tonegawa et al., 2015), indicating these cells "are a population of neurons that are activated by learning". We rather suggest that these neurons that are immature at the time of learning are influencing the memory engram network at the time of retrieval when they are mature (6 weeks-old). We therefore rephrased and clarified the corresponding part of the discussion (page 19)."

2. "Altogether, these results suggest that mature adult-born neurons could sustain encoding and memory expression whereas immature ones at the time of learning would be involved in the stabilization of the trace after reactivation"

Please replace "stabilization of the trace after reactivation" with "stabilization of the trace after behavioral activation".

This is because, as we now all agree, the authors have not tagged a memory trace in the first place.

With these changes, this manuscript is suitable for publication.

REVIEWER COMMENTS

Reviewer #2 (Remarks to the Author):

1. I am glad that the authors found the comments useful and have performed important controls and edits to convey the distinction between reactivation and activation during retrieval. All of that stated, please change title to convey the main finding. The current title "Immature adult-born neurons primed by learning are necessary for remote memory reconsolidation" is too speculative since there is no evidence that the immature neurons are primed during learning and the authors agree**. This is a discussion point and indeed, the authors discuss this idea.

Instead, the title should reflect the main finding: "Adult-born neurons immature during learning are necessary for remote memory reconsolidation". This is the only objective unbiased way forward.

We thank the reviewer for its positive comments. As suggested, we changed the title of the manuscript to "Adult-born neurons immature during learning are necessary for remote memory reconsolidation"

2. "Altogether, these results suggest that mature adult-born neurons could sustain encoding and memory expression whereas immature ones at the time of learning would be involved in the stabilization of the trace after reactivation"

Please replace "stabilization of the trace after reactivation" with "stabilization of the trace after behavioral activation".

This is because, as we now all agree, the authors have not tagged a memory trace in the first place.

We made the change requested by the reviewer #2.

With these changes, this manuscript is suitable for publication.